# Temperature dependence of crystal melt coexistence for supported polyethylene filaments

Da Huang ®[1], Thorsten Hugel ®[2], Bizan N. Balzer ®[2,3] & Günter Reiter ®[1] ✉

An interface or surface may be considered as a planar perturbation reflected by changes in molecular properties in the direction perpendicular to the interface or surface. As a consequence, predicted by theory and shown by experiments, crystals are often covered by a thin liquid layer of their own melt. Such crystal–melt coexistence can be related to phenomena of surface premelting, secondary nucleation and melting point depression, particularly important for small systems. Here, we employed intermittent-contact mode atomic force microscopy imaging on nanoscopic semi-cylindrical filaments of polyethylene on a substrate to observe that these filaments contained a crystalline core bounded by molten regions of rather uniform width, $W_{soft} = (9 \pm 2)$ nm at room temperature, which increased reversibly with temperature $T$. Filaments smaller than ca. $2 \cdot W_{soft}(T)$ were completely molten. The values of $W_{soft}(T)$ compared favorably with theoretically predicted characteristic length scales in the context of nucleation, surface premelting and the melting point depression of finite size crystals. Altogether, we propose that these three phenomena are related and dominated by the intermolecular forces acting at crystal surfaces.

Intermolecular interactions at surfaces, often characterized by the corresponding surface tensions, determine if a solid may be completely or partially wetted by its own melt[1]. In the complete wetting case, a surface starts to melt and forms a thin liquid layer at temperatures $T$ well below the equilibrium bulk melting temperature $T_m^\infty$, a process typically called surface premelting (SM). In the partial wetting case, the solid remains fully crystalline up to and even beyond $T_m^\infty$ (superheating), referred to as surface nonmelting (NM)[2]. Interestingly, as shown for lead[3,4] or gold[2,5], different crystallographic faces of a single substance may show either SM[3,5] or NM[2,4]. For SM, the thickness $l$ of the wetting layer, characterized by the existence of stable interfaces between a crystalline core and a liquid phase at its surface, depends on $T$ and approaches infinity at $T_m^\infty$[3,5,6]. Thus, for a system of finite size, the crystalline core disappears once the liquid phase spans the whole system. This limiting size of the system, where no crystalline phase can

exist, is related to the size of the corresponding critical nucleus[7,8]. Accordingly, neglecting the influence of geometry and curvature, the minimum size of a nanocrystal and the thickness of the layer of its own melt coating the crystal show similar functional dependence on $T$, both length scales grow to large values upon approaching $T_m^\infty$. Related to such crystal-melt coexistence are phenomena of melting point depression[7,8], (secondary) nucleation[9] and surface premelting[3,5,6,10]. In the past, the size- and temperature-dependence of these phenomena have been well studied, showing that the physics behind these three phenomena originates from the broken symmetry of intermolecular interactions at interfaces, typically discussed by the balance of an energy per surface the 〈surface term〉, and an energy per volume the 〈volume term〉. The ratio of 〈surface term〉/〈volume term〉 introduces a length scale $\ell$ which reflects the key physical concept underlying all three phenomena. Several theoretical approaches[11,12] were

[1]Institute of Physics, University of Freiburg, Freiburg im Breisgau, Germany. [2]Institute of Physical Chemistry, University of Freiburg, Freiburg im Breisgau, Germany. [3]Freiburger Materialforschungszentrum (FMF), University of Freiburg, Freiburg im Breisgau, Germany. ✉e-mail: guenter.reiter@physik.uni-freiburg.de

used to describe the temperature-dependence of these phenomena. Experimental investigations[5,6,10,13] and a simultaneous comparison of the resulting observations in terms of nucleation, wetting and melting are rare, in particular for nanoscopic systems.

As shown by computer simulations[14–16] for finite systems in a canonical ensemble $(N, V, T)$, i.e., for a fixed number $N$ of molecules and a finite volume $V$ at a temperature $T$, it is possible to observe stable interfaces between a crystalline and a liquid phase that are in thermodynamic equilibrium. For example, in the absence of an external field or interactions with a substrate, a crystal can be surrounded by a fluid layer[15–20]. As an equilibrium state, such coexistence is stable in time and is re-established even after small perturbations. As further deduced from computer simulations[16], this coexistence does not depend on the geometry of the system and the shape of the crystal. A connection to the size of a critical nucleus in large (infinite) systems has been drawn[16]. While the solid cluster (crystal) found in $(N, V, T)$ simulations of finite systems is in equilibrium with the surrounding liquid, the same cluster in a grand-canonical ensemble $(\mu, V, T)$, where the chemical potential $\mu$, $V$, and $T$ are constant while the particle number $N$ can vary, is unstable and has been identified as the critical nucleus.

In the study presented here, we experimentally explored the coexistence of a nanoscopic polymer crystal with a liquid (molten) layer characterized by a thickness, which we compared with predictions of the Lauritzen–Hoffman theory of secondary nucleation at the crystal growth front, represented by the complementary length scale of the critical nucleus. Our findings propose that, for nanoscopic systems, the temperature dependence of the characteristic length scales of various phenomena, i.e., the size of the critical nucleus, the thickness of the liquid layer in surface premelting, and the size-dependence of the melting-point depression of finite-size crystals, are related. To generate separated polymer crystals of small size, we required a system with a highly limited number of polymer chains. We employed spin-coated ultra-thin films of polyethylene (PE), a well-investigated polymer[21,22]. PE is a nonvolatile polymer, which ensured that the total number of molecules in our sample was a conserved quantity. Using solutions of low polymer concentration, the areal polymer density (represented by a mean film thickness) of these samples was extremely low. Previous experiments[23] have shown that polymer chains in ultrathin layers, often prepared by spin coating[21,23], crystallize very slowly. Sometimes, the crystallization process was so slow that no crystals were detectable within the provided experimental time window of hours or even days[23]. An extremely low nucleation rate and slow surface diffusion caused by attractive interactions with the substrate were invoked as explanations for these observations[24]. In an undercooled melt, crystals can only grow if a heterogeneous or a homogeneous nucleus is established first[24]. However, due to the limited number of polymer chains available in such ultra-thin films, the probability for generating such nuclei is extremely low[24,25]. As a result, typically only very few homogeneous (primary) nucleation events, or none at all, occur within the whole sample within the provided experimental waiting time[23,24].

Fortunately, the homogeneous nucleation step may be circumvented by introducing filaments consisting of stretched and uniquely oriented polymer chains[26,27]. As previously demonstrated[28] through systematic spin coating experiments with dilute polyethylene (PE) solutions, a shear-induced change in polymer chain conformation can occur in the corresponding flow field caused by spin coating, i.e., polymer chains undergo a transition from randomly coiled to stretched chains. As a consequence of the resulting attractive interactions between stretched polymer chains, bundles of PE chains are formed[28–31]. Coalescence and fusion of bundles result in the formation of ultra-long filaments[28]. Due to the massive loss of conformational entropy and the high degree of order of the polymer chains within the filaments (often called "shish")[32], polymer chains may crystallize easily

within the filament[33]. However, during sample preparation, the flow conditions induced by the spinning substrate varied in space and time[28]. Thus, not all polymer chains can undergo such a coil-stretch transition. Consequently, in the resulting as-prepared sample, filaments of stretched and crystalline chains were surrounded initially by non-stretched molten polymer chains.

Filaments can act as sites for secondary nucleation[27,32], circumventing the problem of the low homogeneous nucleation rate of ultra-thin polymer films and allowing the growth of polymer crystals in such films, often resulting in the formation of so-called "shish-kebab"-patterns[27,32,33]. However, to initiate such growth, first a secondary nucleus needs to form on the filament, which requires that the number of available polymer chains is above a threshold value. Furthermore, along the same line of reasoning, we anticipate that crystal growth will stop once the number of remaining molten polymer chains is below this threshold value. For crystals growing on filaments, the polymer chain axis in such crystals is easily identifiable, being identical with the long axis of the filaments. Thus, filaments introduce a unique orientation and define the growth direction for the formation of lamellar polymer crystals, often called "kebabs"[32–34], which will always grow perpendicular to the filament.

A first set of experiments on films of PE was performed at room temperature. This temperature is well below the melting temperature but above the glass transition temperature of PE, allowing for sufficient mobility of non-crystallized polymer chains but also for crystal growth. Furthermore, the adsorption energy of PE on the chosen atomically-smooth and non-epitaxial substrate, i.e., mica, is sufficiently weak not to impede surface diffusion of PE. Thus, all polymer chains had a chance to crystallize. As predicted by theoretical considerations on solid-liquid coexistence[14,15,35], the crystalline core was covered by a layer of molten polymer which was separated from the core by a well-defined interface. To investigate this crystal–melt coexistence, we analyzed the mechanical properties of filaments and their surrounding either crystalline or molten polymer via the phase signal of intermittent-contact mode atomic force microscopy (AFM)[36–45]. Furthermore, we determined the temperature-dependence of the thickness of the liquid layer on a crystal surface for a large range of $T$ below $T_m^\infty$.

## Results

As deduced from their long lifetime of many months, the investigated PE filaments were partially solid. If these filaments had contained molten polymer chains only, they would have decayed into droplets via the Rayleigh-Plateau instability[46]. Naively, one might expect that filaments have uniform mechanical properties of a homogeneous crystalline solid. However, the phase signal derived from the intermittent-contact mode AFM measurements clearly revealed the coexistence of two regions across the filaments, characterized by distinctly different mechanical properties. Remarkably, this observation was independent of the height and diameter of the filaments and similar for both types of studied filaments, as revealed by comparing Fig. 1a and b with Fig. 1c and d. Measurements of the same samples repeated after up to several months of storage at room temperature confirmed that the contrast in the AFM signal for the two regions within the filaments did not change during the time of storage. Thus, we conclude that the observed coexistence of the two regions was stable in time and probably represents an equilibrium pattern.

Clearly, the "hard" region (with a high phase value, thus a high elastic modulus ($E$)) of a filament was bounded symmetrically on both sides by "soft" regions (with a low phase value, thus a low $E$) of similar width (see Fig. 1b and d), labeled as $W_{soft}$. In section 2 of the Supplementary Information we provide more evidence about the coexistence of "soft" and "hard" regions and their identification. From the cross-sections shown in Fig. 1e, we derived the value of $W_{soft}$ as the full width at half maximum of the peak in the phase signal (see Fig. 1e).

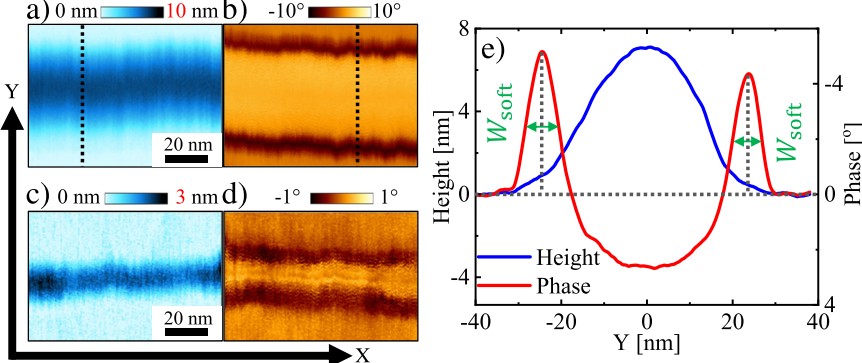

**Fig. 1 | Identifying the soft regions bounding the solid core of a PE filament.**
AFM height (**a, c**) and corresponding phase (**b, d**) images of two different filaments from samples prepared by spin coating a ca. 130 °C hot 0.075 wt% para-xylene solution of PE-M (**a, b**) or a PE-B (**c, d**) onto a ca. 80 °C mica substrate, employing a rotation frequency of 8000 rpm with a 1-second spin-up stage. The set point ratio of the AFM measurement in this figure was $s = 0.995 \pm 0.005$. For an easier comparison of (**a**) with (**b**) and (**c**) with (**d**), the phase images here were mirrored horizontally (i.e., at the Y-axis). For the dashed lines indicated in images (**a**) and (**b**), the corresponding height $\hat{H}(x, y, f_\mathrm{T}, E)$ and phase cross-section data are shown in (**e**). There, the characteristic width ($W_\mathrm{soft}$) of the soft region (indicated by green arrows) is defined as the full width at half maximum of the peak in the phase data. A more detailed discussion on measurement and definition of $W_\mathrm{soft}$ can be found in Section 1 of the Supplementary Information.

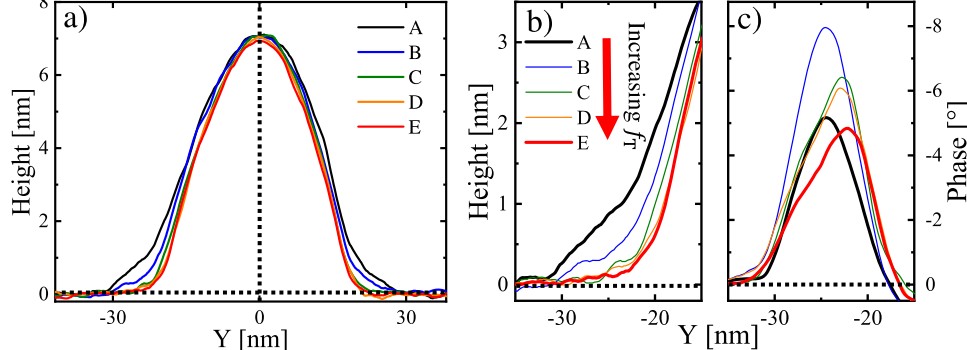

**Fig. 2 | Examining the properties of the soft regions. a** Height cross-sections taken at the position indicated by the dashed line in Fig. 1a. The corresponding increase in force $f_\mathrm{T}$ caused a stronger deformation, especially in the soft part at the boundary of the filament. Focusing on this soft region, we highlight these changes in height (**b**) and the corresponding changes in the phase values (**c**) at varying set point ratio $s$, decreasing in step of 0.01 from $s = 0.995 \pm 0.005$ to $s = 0.955 \pm 0.005$ for A to E, respectively. When the cantilever tip pierced through the whole thickness of the soft region (indicated by the region of almost zero height in curves C to E shown in (**b**)), the interaction with the underlying hard material (mica substrate or solid PE crystal) is more and more dominating the phase signal, reflected by a gradual approach of the phase values measured on the substrate. Furthermore, for the smallest values of the force $f_\mathrm{T}$, surface tension is strongly hindering penetration, yielding somewhat smaller values of the phase signal than the ones observed when the cantilever tip penetrated the soft layer.

Furthermore, $W_\mathrm{soft}$ was found to be rather constant along the whole filament. For all studied PE filaments and independent of their size, the coexistence of a hard (solid) core bounded by a soft (liquid-like) layer near the filament-substrate contact line was confirmed. We notice that, for different geometries like lamellar crystals of highly folded crystalline states of polymers, further dedicated investigations need to be performed.

We performed systematic AFM measurements on a single PE filament using intermittent-contact mode with a decreasing setpoint value, thus a decreasing setpoint ratio ($s$)[44] (the definition and details of $s$ are described in the "Methods" section.) Depending on the force ($f_\mathrm{T}$) applied via the AFM tip onto the sample, the AFM cantilever tip penetrates the material on the sample surface[38,43,45]. For small values of $f_\mathrm{T}$, the depth of penetration ($d_\mathrm{p}$) at position $(x, y)$ into the sample also depends on $E$ of the sample[36]. Thus, $d_\mathrm{p}$ is recorded as $d_\mathrm{p}(x, y, f_\mathrm{T}, E)$. In Fig. 1a, the directions of $x$ and $y$ are defined. For a given AFM cantilever tip with a fixed shape, for small values of $f_\mathrm{T}$ and for a sample with time-independent mechanical properties, the measured deformation of the sample increases linearly with increasing $f_\mathrm{T}$[40]. Deformation is often visualized as a change in topographical features of the sample, expressed by changes in the measured height data. As a result, the measured surface topography of the sample[37,43,45], expressed through the apparent height $\hat{H}(x, y, f_\mathrm{T}, E)$ at position $(x, y)$, which will deviate from the true surface topography $H(x, y)$ of the sample by the depth of penetration, i.e., the value $d_\mathrm{p}(x, y, f_\mathrm{T}, E)$. Therefore, we get: $H(x, y) = \hat{H}(x, y, f_\mathrm{T}, E) + d_\mathrm{p}(x, y, f_\mathrm{T}, E)$. By decreasing $s$ at fixed other experimental parameters, $f_\mathrm{T}$ increased.

For the series of AFM measurements shown in Fig. 2, we used the same AFM cantilever tip. Because the gain values of the AFM were not changed during a set of measurements, for the small range of values of $s$ investigated, we can assume that $f_\mathrm{T}$ increased linearly with the decrease in $s$, i.e., $f_\mathrm{T} \sim 1 - s$[41]. Furthermore, upon increasing $f_\mathrm{T}$, also $d_\mathrm{P}(x, y, f_\mathrm{T})$ increased[40,43,45]. Thus, we observed a decrease in $\hat{H}(x, y, f_\mathrm{T})$, which was larger in the soft regions than in the hard regions. (Fig. 2a, b). For the conditions applied here, in particular for the small values of $d_\mathrm{P}$, the area of contact between tip and sample is expected to vary only slightly. Therefore, in parallel to the increase in $f_\mathrm{T}$, we also have an increase of the applied pressure.

Besides, once the AFM cantilever tip pierced through the whole thickness of the soft region, i.e., yielding eventually $\hat{H}(x, y, f_\mathrm{T}) \cong 0\,\mathrm{nm}$ in this soft region, the correspondingly observed phase value was progressively dominated by the properties of the underlying mica

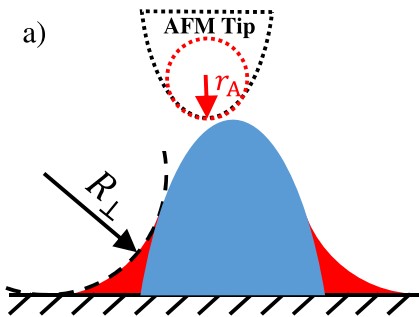

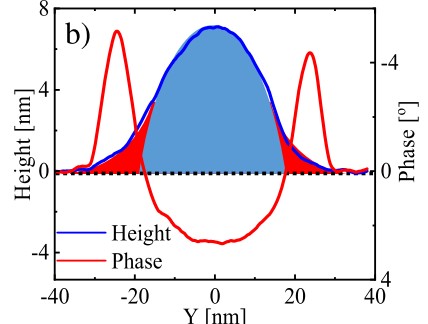

**Fig. 3 | Interpretation of the measured properties. a** Cross-sectional sketch of a filament with a hard core (blue) bounded by soft regions (red). We represent the soft regions as concave menisci (with a radius of curvature $R_\perp$) and the hard core, consisting of bundles of stretched PE chains, through a parabolic shape, approximating the convolution of the filament with the shape of the AFM cantilever tip, characterized by its radius of curvature $r_A$. (**b**) Superposition of the sketch in (**a**) with the measured cross-sections of the AFM height and phase data of the sample shown in Fig. 1e, results in a satisfying match.

substrate (which, in our case, has been set to a phase value of 0 degrees). Accordingly, upon increasing $f_T$, the phase value of the soft region ultimately approached the value of the mica substrate (see Fig. 2c).

The filaments investigated here consisted only of a single polymer, i.e., polyethylene (PE). All measurements shown in Figs. 1, 2 were performed at the same temperature, i.e., at room temperature (ca. 20 °C), which is between the glass transition temperature and the melting temperature of PE. Thus, the large difference in the values of the intermittent-contact mode phase signal demonstrates that, even at room temperature, within each PE filament, a solid crystalline core coexisted with meniscus edges of molten polymer, i.e., soft zones at the lateral boundary of the filament.

As shown in Figs. 1, 2, and sketched in Fig. 3, filaments were covered by a region of soft molten polymer. Here, we consider filaments as long semi-cylinders on a planar substrate. Due to the semi-cylindrical geometry of the nanofilament and its interaction with the wettable mica substrate, we have to deal with different values of the interfacial energies and surface tensions across the filament. One possibility is that not all parts of the filament are covered with a liquid layer, and the conditions for surface melting (or non-melting) are not fulfilled for all parts of the nanofilament. The other possibility is that an ultrathin liquid layer exists at the top of the filament. In the latter case, it is so thin that its AFM phase signal is dominated by the mechanical response of the underlying crystalline material, possibly with a minor (insignificant) contribution from this ultrathin surface liquid layer. In contrast, the meniscus region exhibits a much greater effective thickness. There, the AFM phase response is clearly dominated by the properties of the molten polyethylene—at least until the tip fully penetrates this region. Thus, the observed soft regions in the AFM phase images correspond to the molten polymer chains in the concave meniscus region at the edges. In section 2 of the Supplementary Information, we provide a set of force spectroscopy measurements taken at successive positions along the cross-section of a filament, supporting the sketch shown in Fig. 3a.

We approximated the cross-section of the soft region by a concave shape (see Fig. 3a). The concave outline of this wedge-shaped meniscus region is favoring capillary condensation, as reflected through the Kelvin equation[47–49]. Thus, as a consequence of the acting capillary forces, polymer chains accumulated in the wedge, forming a liquid-like zone near the (filament-substrate-air) three-phase-contact-line. The number of polymer chains which potentially could diffuse to the wedge-shaped meniscus region depended on the number density of polymer chains adsorbed on the substrate surrounding the filament. The surface of the meniscus region is characterized by a constant radius of curvature $R_\perp$. We note that the curvature in the direction along the filament is approximately zero (i.e., $R_\parallel \approx \infty$). Accordingly, as a consequence of the mean curvature $\bar{R} = \left(\frac{1}{R_\perp} + \frac{1}{R_\parallel}\right)^{-1} \approx R_\perp$, the pressure $p_{wedge}$ inside this wedge of molten polymer was lower than the surrounding atmospheric pressure $p_{atm}$, i.e., the Laplace pressure was negative, $\Delta p_{Laplace} = p_{wedge} - p_{atm} < 0$. A negative Laplace pressure, which was constant ($\bar{R} = $ constant) within the entire soft region along the whole filament, assured that thickness fluctuations were dampened. Thus, within some small fluctuations, the height and the width of the soft region remained constant along the whole filament.

The fact that we have observed on the same spin-coated sample two types of straight filaments, some with a uniform width and others decorated with "beads-on-a-string" patterns, suggests the existence of a threshold number of polymer chains that can be included in the meniscus region. Beyond this threshold value, secondary nucleation of crystalline structures becomes possible. Such nucleation events lead to the formation of so-called "shish-kebab" crystalline patterns, characterized by edge-on lamellar crystals growing perpendicular to the long axis of the filament. As demonstrated previously for this type of sample preparation based on spin coating, filaments containing stretched polymer chains are co-deposited together with non-stretched polymer chains. The latter forms an adsorbed ultrathin layer of liquid-like polymer in the surroundings of the filaments. We assume that filaments acted as attractors for the surrounding molten non-stretched chains. Eventually, these chains form a liquid meniscus-shaped zone around the filaments. We further assume that, for a width larger than $W_{soft}$, these soft regions become unstable. Possibly caused by a kind of Plateau–Rayleigh-type instability, the liquid meniscus region is transformed into many bead-like structures on the filaments, thereby generating a "beads-on-a-string" pattern. Given that the size of the beads is larger than the critical size of a crystalline nucleus, the beads partially crystallized. Deduced from the corresponding intermittent-contact mode phase signal, these beads also consisted of a hard core bounded by a soft region. In section 3 of the Supplementary Information we provide AFM images for filaments with beads-on-a-string patterns.

Given that the surface of the core of a filament (often called shish) consisted of locally ordered, crystalline chains, epitaxy may allow for the induction of secondary nucleation followed by the growth of edge-on lamellar crystals (often called kebabs). However, formation of kebabs is only possible if a sufficient number of polymer chains is available at its growth site. A secondary nucleus requires a certain number of crystalline stems of a certain length which can be derived from a single or multiple polymer chains[50,51]. We note that an analogy but also some differences exist to observations of substrate-induced prefreezing of polymer chains, for example, on graphite[19,20]. In both cases, we observe stable coexistence of crystalline and molten polymer

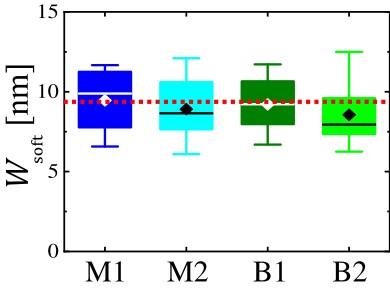

**Fig. 4 | $W_{soft}$ is independent of size and morphology of PE filaments.** Box plot of values of $W_{soft}$ for two types of samples, prepared by spin coating a ca. 130 °C hot 0.075 wt% para-xylene solution of PE-M (label M) and a PE-B (label B), respectively, onto a ca. 80 °C mica substrate using 8000 rpm with a 1-second spin-up stage. $W_{soft}$ was derived from uniform (straight) filaments of variable height and width (M1, dark blue; B1, dark green) or from thicker bead-like structures on filaments with a beads-on-a-string morphology (M2, light blue; B2, light green), respectively. In all four cases, $W_{soft}$ has a mean value of approximately 9 nm, as indicated by the red dotted line. All data points resulted from multiple measurements on filaments of different sizes; all were performed at room temperature. The set point ratio $s$ of all these AFM measurements varied between $0.985 \pm 0.005$ and $0.995 \pm 0.005$. Each box plot has a data pool obtained from around 20 measurements. The box plots in this paper are constructed using the 0th, 25th, 75th, and 100th percentiles and include all data points without excluding outliers. The solid line inside the box indicates the median value; the diamond indicates the mean value. In section 4 of the Supplementary Information, we provide AFM phase images for filaments and beads-on-a-string structures measured with two kinds of cantilevers, having different spring constants, demonstrating that the resulting values of $W_{soft}$ were independent of the cantilevers used.

chains. In the case of prefreezing, the thickness of a crystalline layer close to the substrate is governed by polymer-substrate interactions. This crystalline layer can span the whole substrate and has a melting temperature above the equilibrium melting temperature of the bulk polymer[19,20]. For the filaments investigated here of finite size, a solid central part was coexisting with soft boundary zones of a fixed width and a distinctive temperature dependence which is discussed later.

For a quantitative statistical analysis of $W_{soft}$, we have performed measurements on a multitude of filaments, including filaments with a beads-on-a-string morphology. Furthermore, we prepared filaments from solutions of the two different PE polymers (PE-M and PE-B) investigated in this study. The summary of our statistical results for $W_{soft}$ is shown in Fig. 4. It is highly intriguing that for all these measurements at room temperature, taken on filaments of various heights and differing in morphology, $W_{soft}$ was almost identical at a value around 9 nm, with a rather narrow spread of the results.

In the following, we examine the origin of the coexistence of molten and crystalline polymer chains in such nanoscopic filaments adsorbed on a wettable substrate based on the Lauritzen–Hoffman theory of secondary nucleation at the growth front of the crystal. This elementary assumption is stimulated by the observation that bead-like structures (kebabs) were formed only when sufficient molten polymer chains were available. Crystallization is only possible for volumes larger than $V_{cr}$. For the derivation of the size of the critical nucleus, we employ classical nucleation theory (CNT). For nucleation within the soft zone at the boundary of the filament, we assume, following the approach used in reference [9], a cylindrical critical nucleus which has the size of $V_{cr} = \pi r^{*2} \cdot l^{*}$. There, $r^{*}$ is the radius and $l^{*}$ is the length of the cylinder[9].

PE crystals grow by secondary nucleation at the crystal growth front. In order to form a stable (critical) secondary nucleus, a minimum volume of molten polymer chains needs to be available at the growth front of the crystal[50]. However, in finite systems like ultra-thin nanometer-sized filaments, the number of available molten polymer chains is easily exhausted. Thus, once the number of molten polymer

chains in the surrounding and at the growth front is below this minimum value, we anticipate that crystal growth will stop. Accordingly, if the residual number of molten polymer chains at the growth front per appropriate length along the filament adds up a total volume less than $V_{cr}$, these polymer chains cannot crystallize and will stay in the molten state. Moreover, since the secondary nuclei adopt the same orientation as the original crystal—that is, the nanofilament in which the polymer chains are aligned along its axis—the cylindrical nucleus is aligned lengthwise along the nanofilament, while its radius is oriented perpendicularly; this radius is directly correlated with the value of $W_{soft}$, i.e., $W_{soft} \sim r^{*}$. This implies that the crystal-melt coexistence observed at the growth front of the polymer crystal differs from the well-studied coexistence at the fold surface. Consequently, we anticipate that filaments with a crystalline core (blue area in Fig. 3a) are surrounded with a narrow zone of molten polymer at the growth front (red area in Fig. 3a). In order to support our interpretation, we have superposed the sketch of Fig. 3a with the measured cross-sections of the AFM height and phase data shown in Fig. 1e (see Fig. 3b). There, the phase signal switches from negative values at the periphery of the cross-section to a positive value in the center of the filament, representing the soft zones of molten polymer at the periphery and the hard (crystalline) core of the filament, respectively.

Invoking the concept of secondary nucleation for the soft regions at the periphery of the filament and employing CNT, we arrive at two predictions for the equilibrium width $W_{soft}$ of the soft zone of molten polymer. First, at a given temperature, employing thermodynamic arguments inherent to CNT, we assume that a cylindrical secondary nucleus can only form in the soft zone of molten polymer at the periphery of a hard core of a filament when the width $W_{soft}$ of this zone is larger than the size of the critical secondary nucleus, characterized by $r^{*}$, i.e., we assume $W_{soft} \sim r^{*}$. For fixed experimental conditions, we expect that the size $r^{*}$ of the critical secondary nucleus is constant and should not depend on crystal size or morphology, i.e., filaments having either a uniform width or being decorated with beads-on-a-string patterns should yield the same value of $W_{soft}$. In fact, all values of $W_{soft}$ shown in Fig. 4 were similar and narrowly distributed around $(9 \pm 2)$ nm. Accordingly, within error bars, the observation of a constant value of $W_{soft}$ is consistent with the assumption of $W_{soft} \sim r^{*}$.

For long polymer chains, $r^{*}$ is independent of polymer molecular weight[52]. Thus, the dispersity of the polymer is not expected to influence either $r^{*}$ or $W_{soft}$. Furthermore, for samples prepared on the same type of substrate (here: mica), i.e., for identical wetting conditions related to the surface tension of the substrate and the polymer, we may assume that the geometry of the soft meniscus region containing molten polymer is the same for all filaments, independent of their size or crystal morphology. In fact, for the two PE samples with largely different dispersity in chain length, our experimental results yielded similar values of $W_{soft}$.

The second prediction based on the concept of secondary nucleation concerns the temperature dependence of $W_{soft}(T)$, which is expected to be proportional to $r^{*}(T)$. According to CNT, the temperature dependence of $r^{*}$ can be derived from the following equation[9]:

$$r^{*} = \frac{2\sigma_s}{\Delta G_V} \quad (1)$$

where $\Delta G_V$ is the free energy change, per unit volume, involved in the formation of a crystal nucleus, $\sigma_s$ is interfacial or surface energy of the nucleus.

$\Delta G_V$ can be approximated by ref. 9:

$$\Delta G_V \approx \rho_m \cdot \Delta H_f \cdot T \cdot \Delta T / (T_m^{\infty})^2 \quad (2)$$

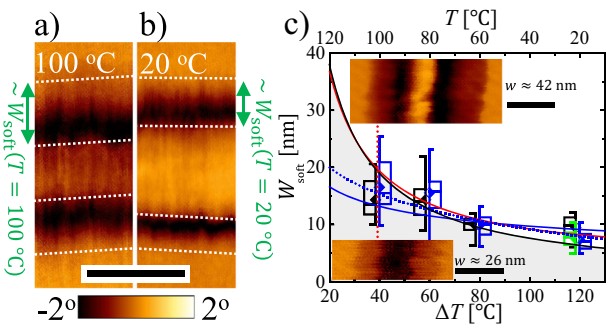

**Fig. 5 | Temperature dependence of $W_{soft}$ and its relation to secondary nucleation, surface premelting, and size-dependent melting point depression.** The PE-M sample was used here. (**a, b**) AFM phase images taken at different temperatures at the same position of a filament. In (**a**), the sample was measured after heating to $\approx 100\,°C$, i.e., $\Delta T = T_m^\infty - T \approx 41\,°C$. In (**b**), the sample was measured after being cooled back to room temperature ($\Delta T \approx 120\,°C$). For an easier comparison, image (**b**) was mirrored horizontally. The dotted white lines in (**a**) and (**b**) indicate the boundaries of the soft regions. (**c**) Statistical values of $W_{soft}(\Delta T)$ derived from AFM measurements performed at different temperatures. The data point in green color denotes the value first measured at room temperature before heating the sample to 100 °C. The symbols are split in a black (green) and a blue part, representing values of $W_{soft}$ obtained from straight filaments and from filaments decorated with bead-like structures, respectively. Each box plot represents the average from a pool of around 20 measurements. This box plot employs the same methodology as Fig. 4. The red line and the blue lines reflect the theoretically predicted temperature dependence based on CNT (equation (**a**) in Table 1) and surface premelting (dotted: equation (**b2**) in Table 1, and solid: equation (**b1**) in Table 1), respectively, adjusted to the measured values of $W_{soft}$. The black line denotes the Gibbs-Thomson equation (equation (**c**) in Table 1) which also characterizes the transition from fully molten filaments to filaments with a coexisting crystalline core as a function of the width $w$ of the filaments. More details are given in the main text. The two AFM phase images were measured at $T \approx 100\,°C$, indicated by the vertical dotted red line, for filaments which have a width $w \approx 26\,nm < 2 \cdot W_{soft}$ (bottom image) and $w \approx 42\,nm > 2 \cdot W_{soft}$, respectively. The color code ranges from $-5°$ (dark) to $5°$ (bright) for both images. In this figure, all AFM data were measured using $s = 0.990 \pm 0.005$ on the same sample as Fig. 1a. All scale bars in this Figure represent 20 nm.

There, $T$ is the experimental temperature of the sample (filament), $\Delta T = T_m^\infty - T$, $\rho_m$ is the mass density of polymer, $\Delta H_f$ is the heat of fusion per unit mass of the polymer at $T_m^\infty = 141\,°C$.

Following refs. 53,54, we used the following values for PE: $\sigma_s$ ranges from $12\,mJ/m^2$ to $15\,mJ/m^2$; $\Delta H_f \approx 4.1 \pm 0.2\,kJ/mol$ of $CH_2$, and $\rho_m$ is ranging from ca. $780\,kg/m^3$ to ca. $970\,kg/m^3$. Based on Eq. (2), we obtain $r^*(T)$ or analogously $r^*(\Delta T)$:

$$r^*(\Delta T) \approx P \cdot \frac{(T_m^\infty)^2}{T \cdot \Delta T} \qquad (3)$$

where $P = \frac{2\sigma_s}{\rho_m \cdot \Delta H_f} = (0.10 \pm 0.02)\,nm$. Furthermore, as a consequence of the thermodynamic arguments underlying CNT, changes of $r^*$ induced by varying $T$ are expected to be reversible. Based on the relation $W_{soft} \sim r^*$, we anticipate that $W_{soft}(T)$ should become larger at higher temperatures and should become smaller again when $T$ was reduced back to lower temperatures, e.g., to room temperature.

To support our hypothesis of a relation between secondary nucleation and the measured values of $W_{soft}(T)$, we performed a series of AFM measurements at different temperatures $T$. We employed the following sequence of temperatures: Room temperature (ca. $20\,°C) \rightarrow 100\,°C \rightarrow 80\,°C \rightarrow 60\,°C \rightarrow$ back to room temperature. As can be seen by comparing Fig. 5a, b, the width of the soft zone was clearly larger at the higher $T$. From a series of experiments, the statistical

**Table 1 | Functional dependence of the temperature variation of the characteristic length scale, represented as $\ell$, of a liquid layer on the surface of a crystal, as predicted by three different phenomena**

| | | |
|---|---|---|
| Classical Nucleation Theory[9] | $\ell \sim (T\Delta T)^{-1}$ | (a) |
| Surface Premelting[6,11,13] | $\ell \sim \Delta T^{-1/3}$ | (b1) |
| | $\ell \sim -\ln\Delta T$ | (b2) |
| Gibbs-Thomson equation[7] | $\ell \sim \Delta T^{-1}$ | (c) |

All relationships disregard constant prefactors and additive constants. More details are given in the main text.

$\ell$ is the ratio of ⟨surface term⟩/⟨volume term⟩, which is at the basis of the size of the critical nucleus, reflects the thickness of the liquid layer in surface premelting, and the length scale that controls the amount of melting point depression. $T$ is the temperature of the sample, $\Delta T = T_m^\infty - T$, where $T_m^\infty$ is the equilibrium bulk melting temperature of the sample.

analysis of $W_{soft}$ showed a decrease as a function of increasing degree of supercooling $\Delta T$. On a multitude of samples, measurements were performed on various filaments, with either a uniform width or they were decorated with beads-on-a-string patterns. Most importantly, this increase of $W_{soft}$ with $T$ was fully reversible as demonstrated by the values of $W_{soft}$ obtained at room temperature, measured before and after heating the sample to 100 °C, represented by the black and green data points in Fig. 5c. In section 5 of the Supplementary Information we provide additional AFM images supporting the reversibility of the changes of $W_{soft}(T)$.

When comparing $r^*(\Delta T)$, as given by Eq. (3), with the experimentally measured values of $W_{soft}(\Delta T)$, because we have to account for the specific geometry of the wedge-shaped soft zone of molten polymer and the influence of the mica substrate, $W_{soft}(\Delta T) \neq r^*(\Delta T)$, but only proportional to $r^*(\Delta T)$, i.e., $W_{soft}(\Delta T) \sim r^*(\Delta T)$. Thus, a constant prefactor $C_{CNT}$ is required to relate $W_{soft}(\Delta T)$ with $r^*(\Delta T)$, we obtain $W_{soft}(\Delta T) = C_{CNT} \cdot r^*(\Delta T)$. Using $C_{CNT} = 16$ and employing Eq. (3) for $r^*(\Delta T)$ yields the red line shown in Fig. 5c. A good match with the experimentally determined data points is observed. Thus, we can conclude that all filaments, including filaments decorated with bead-like structures, were bounded by coexisting soft zones of molten polymer of a size just below a size proportional to $r^*(\Delta T)$ of the critical secondary nucleus.

Besides the limiting size derived from the minimum volume of molten polymer required for the formation of a stable (critical) secondary nucleus at the crystal growth front, the coexistence of a liquid layer on a crystal is also theoretically predicted[6,11,13] and experimentally verified[6,11,13] for planar surfaces on bulk samples in the context of surface premelting. Assuming a specific distance-dependence of intermolecular interactions, various functional forms of the temperature variation of the characteristic length scale, represented as $l$, of a liquid layer on the surface of a crystal have been predicted (see Table 1). However, these approaches did not account for the influence of geometry, curvature, and the influence of the mica substrate, making a direct comparison with our results problematic. Nonetheless, as seen by the blue lines in Fig. 5c, within the uncertainty of our data points of $W_{soft}(\Delta T)$, a description along the lines of surface melting is possible.

From Fig. 5, we can deduce that the increase of $W_{soft}$ with $T$ indicates the progressive melting of PE crystals at their surface, suggesting that melting of PE crystals is a continuous process[55]. Interestingly, for a given finite size of a filament, characterized by its width $w$, such an increase of $W_{soft}(T)$ allows to define a size-dependent melting temperature of a filament $T_m(w) \leq T_m(w=\infty) = T_m^\infty$, a phenomenon often referred to as size-dependent melting point depression of nanostructures[7,8]. In other words, when $2 \cdot W_{soft}(T) \approx w$, then the whole filament is molten. Accordingly, thinner (narrower) filaments melt at lower temperatures than wider ones. The corresponding temperature dependence of $T_m(w) \sim \Delta T^{-1}$ is described by the Gibbs-

Thomson (GT) equation[7,56] (see Table 1). Some typical AFM phase images in support of the melting point depression in nanoscopic filaments are shown as insets in Fig. 5c. The GT-equation is represented by the black line in Fig. 5c, which can be interpreted as a phase boundary between the region where nanocrystals do not exist – they are completely molten [for $w \lesssim 2 \cdot W_{soft}(T)$]– and the region where (nanoscopic) crystalline core coexists with a molten surface layer [for $w \gtrsim 2 \cdot W_{soft}(T)$]. In section 5 and 7 of the Supplementary Information, we provide additional AFM images in support of the melting point depression in nanoscopic filaments. As shown previously[57], even after the complete melting of the nanoscopic filaments, the stretched polymer chains inside the filament kept some memory which decayed only slowly.

As described and discussed above, we can relate our experimental observations of crystal−melt coexistence and the corresponding temperature dependence of $W_{soft}$ with three different phenomena and the corresponding characteristic length scales: the size of the critical secondary nucleus, the thickness of the molten wetting layer and the size-dependent melting point depression of nanostructures. Although these phenomena were often treated independently, our results highlight their fundamental relationship. All three phenomena originate from the broken symmetry of intermolecular interactions at surfaces and the corresponding competition between interfacial energy and internal energy.

## Discussion

Exploiting the high sensitivity of intermittent-contact mode AFM for distinguishing even small differences in elastic properties of polymer, we have been able to visualize the coexistence of soft and hard regions within ultrathin filaments of polyethylene deposited on an atomically smooth mica substrate, a highly controllable system of simple geometry. For all AFM measurements performed at room temperature, a soft region of rather uniform width $W_{soft}$ = (9 ± 2) nm was observed near the contact line of the polymer filament with the substrate. We note that polyethylene has a high equilibrium melting temperature of $T_m^\infty = 141\,°C$ but a glass transition temperature below room temperature. Therefore, we conclude that the soft region contained molten polymer which bounded the solid core region of the filament consisting of the crystallized polymer. Our measurements proved that, for a given temperature, the value of $W_{soft}$ did not depend on the size (width and height) of the filament. Even filaments decorated by substructures like beads-on-a-string patterns showed the same value of $W_{soft}$. Upon increasing the temperature $T$ towards $T_m^\infty$, $W_{soft}(T)$ increased in size. These changes of $W_{soft}$ with $T$ were fully reversible. Furthermore, for filaments with a width $w \lesssim 2 \cdot W_{soft}(T)$, the whole filament was molten.

We compared our measured values of $W_{soft}(T)$ with the size of the critical secondary nucleus required for the growth of polymer crystals, the theoretically predicted values of the thickness of the wetting layer (surface premelting) on a planar surface, as well as the corresponding melting point depression of finite size crystals. We note that our approach allowed to probe all three phenomena simultaneously. If the size of the soft zone of molten polymer, characterized by its width $W_{soft}(T)$, was less than the required minimum for generating a stable secondary nucleus of critical size, crystallization did not occur. Analogous to the temperature dependence of the size of the critical secondary nucleus, $W_{soft}(T)$ increased in size upon approaching the bulk melting temperature $T_m^\infty$. In parallel, the width of the core region decreased with increasing temperature. Thus, the comparable temperature dependence of the size of the critical secondary nucleus and the one of $W_{soft}(T)$ indirectly confirms that the core of the PE filament (resulting from bundles of stretched chains) was in a crystalline state as long as its width was larger than the corresponding size of the critical nucleus. While the physical concepts underlying these three phenomena are different, they all were capable to describe our results of

$W_{soft}(T)$ within the uncertainty of the datapoints, highlighting the fact that all these phenomena originate from the same intermolecular interactions at the surface of crystals.

## Methods

### Materials and sample preparation

For the present study, we employed long polymer filaments of PE obtained by spin coating dilute solutions of PE in para-xylene (PX, Sigma-Aldrich)[28]. Most filaments were prepared from a 0.075 wt% dilute solution of PE in PX, which was heated to ca. 130 °C in a thermostat. This solution was spin coated onto a ca. 80 °C hot and freshly cleaved mica substrate (Plano GmbH, Wetzlar/Germany) rotating at 8000 rpm (using an acceleration period of 1 second for the spin-up stage). Two different PE samples were examined: PE-M, consisting of PE with a weight-average molecular weight $\overline{M_W}$ = 1800 kg/mol and a dispersity Đ = 3.1 (Institut für Makromolekulare Chemie, Freiburg/ Germany), and PE-B, representing a blend of 50 wt% of PE-M and 50 wt % PE with $\overline{M_W}$ = 277.3 kg/mol and a Đ = 27.4 (Sinopec Research Institute of Petroleum Processing). Spin coating experiments were performed with a device from Laurell Technologies Corporation, Lansdale, PA, USA (Model WS-650MZ-23NPP).

### Characterization techniques

Topography and mechanical properties of all samples were determined with the help of an atomic force microscope (AFM, NanoWizard II from JPK Instruments, Berlin/Germany). For all measurements, operated in the intermittent-contact mode in air, we used Tap190Al-G AFM cantilever tips from BudgetSensors, Bulgaria with a characteristic radius of curvature of around 10 nm, a characteristic spring constant of the cantilever of around 48 N/m, and a resonance frequency of the cantilever of around 165 kHz. The scan angle did not have any detectable influence on the resulting images. The values of the phase-signal derived in the AFM intermittent-contact mode allowed to detect differences in elastic properties of the measured sample[36,39,42,43,45]. For our AFM measurements, which were performed in the repulsive regime, we chose a parameter setting where a lower phase value referred to a lower elastic modulus[36]. For both, phase and height data, the value characterizing the surrounding mica substrate (the background) was used as a reference, which was set to a value of zero. Therefore, the "soft zone" is defined as the area with a negative phase value, high energy dissipation; the "hard core" of filaments is defined as the area with a positive phase value, i.e., low energy dissipation[36,42,58].

For the in-situ high-temperature AFM measurements, we used a High-Temperature Heating System (HTHS™) from JPK Instruments, which we calibrated with a K-type thermocouple before each measurement. To allow for thermal equilibration after each step-wise change in temperature, the sample was kept at the set temperature for more than 30 minutes before starting the measurement.

During a typical intermittent-contact mode AFM measurement (in air), the AFM cantilever oscillates at a frequency, which is in the order of 100 Hz lower than one of the resonance frequencies, with a "free" amplitude $A_0$, obtained at a position where the cantilever tip was at least 100 μm from the sample surface[36,59]. When the cantilever approaches the sample surface so closely that attractive van der Waals interactions between AFM cantilever tip and sample start to become significant, the oscillation amplitude may increase, but subsequently, at even closer distances to the surface, will decrease below $A_0$ due to repulsive interactions and some deformation of the sample induced by the force applied through the cantilever. The force applied to the sample surface is proportional to the spring constant and the deflection of the cantilever. An electronic feedback loop (in this paper, I-gain is 0.02, P-gain is between 30 Hz to 50 Hz) acts on the piezoelectric scanner of the AFM, providing precise control over the oscillation amplitude of the cantilever. During scanning of the AFM cantilever tip

**Article** https://doi.org/10.1038/s41467-025-67465-2

across the sample, the movement of the piezoelectric scanner in the vertical direction is adjusted in order to keep the oscillation amplitude of the cantilever at a pre-set value $A$, the so-called set point, that is slightly below the oscillation amplitude ($A_0 = 73 - 80$ nm)[38]. (A more detailed discussion about the estimation of $A_0$ can be found in Section 6 of the Supplementary Information) Therefore, in this constant amplitude mode, AFM images were acquired at a constant value of the set point ratio $s$, with $s = A/A_0$.

For thin layers and topographical features with small values $H(x, y)$ on a substrate of high elastic modulus, we have to avoid that the penetration depth $d_p$ is larger than $H(x, y)$ or the layer thickness. Thus, in the here presented experiments, we have chosen the lowest experimentally accessible forces, i.e., we have chosen values of the set point ratio close to $s = 1$[38].

## Data availability
The findings of this study are fully documented by the data provided in the manuscript and the supplementary information. The plotted and statistical data generated in this study have been deposited in the supplementary Excel file entitled: Original data-crystal melt coexistence. All data are available from the corresponding author upon request. Source data are provided with this paper.

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

## Acknowledgements

The authors thank Dr. Toshikazu Miyoshi (三好 利一) for inspiring and fruitful discussions. T.H. and B.B. thank the Deutsche Forschungsgemeinschaft (DFG, German Research Foundation) under Germany's Excellence Strategy – EXC-2193/1 – 390951807 for support.

## Author contributions

D.H. and G.R. conceived the idea for this work and designed the experiments. D.H. performed all experiments. B.B. performed control experiments and calibrated the spring constant of the AFM cantilever (PPP-NCSTR), which was used for the force spectroscopy measurements. D.H., B.B., T.H., and G.R. discussed the results, commented on the interpretation and conclusions presented in the paper, and contributed to writing and revision of the paper.

## Funding

## Competing interests

The authors declare no competing interests.
