## [Transparent Peer Review file · Nature Communications]

Temperature dependence of crystal melt coexistence for supported polyethylene filaments

Corresponding Author: Professor Günter Reiter

Version 0:

Reviewer comments:

Reviewer #1

(Remarks to the Author)
Review attached.

[Editorial Note: See end of file]

Reviewer #2

(Remarks to the Author)
Please see the attached Word file with my 3.5-page report, which includes symbols and subscripts that are not transferred correctly when I paste the report into this window.

[Editorial Note: See end of file]

Reviewer #3

(Remarks to the Author)
This paper presents an interesting study on the coexistence of a crystal- melt interface on supported polyethylene filaments.

I found that the paper is not easy to read, and at times it is confusing. I urge the authors to revise it and make it more understandable.

The first phrase of the abstract is quite cryptic and not well written: "

The broken symmetry of molecular interactions at interfaces is causing that crystals are often covered by a thin liquid layer of their own melt." No explanation is given to what they refer to as broken symmetry and they cannot use the verb in this tense "is causing", as this denotes that is happening at this instant...Consider revising and rewriting the abstract to indicate clearly what has been done and what is new and a significant advancement with respect to existing literature.

The write: For all studied PE filaments and independent of their size, the coexistence of a hard (solid) core bounded by a soft (liquid-like) layer near the filament-substrate contact line was confirmed.

Are they sure this holds for any PE filament size? Are we to expect a molten surface in a high molecular weight microscopic-sized PE fiber or bundles of fibers to have the same molten surface.

It would seem that the phenomena observed is most likely present in nanometer fibers.

In many instances, the authors use the word "polymer" to refer to polymer chains, for instance, just one example: "However, in finite systems like ultra-thin nanometer-sized filaments the number of available molten polymers is easily

exhausted. Thus, once the number of molten polymers in the surrounding and at the growth front is below this minimum value, we anticipate that crystal growth will stop."

There is no way they can use the expression "number of available molten polymers is easily exhausted." I imagine they are referring to the available molten chains; otherwise, I do not understand their meaning. This happens several times in the paper.

The experiments performed and the analyses presented are very good, but unfortunately, the paper is in many parts not clear enough, and the writing should be improved.

They have employed only PE filaments, but in principle, they could have used other filaments where they can easily vary the degree of crystallinity and check its influence (i.e., PET).

The intermolecular interactions in this case are weak Van Der Waal forces (as only PE was employed). What would happen if a polymer like PEO is used or PCL, in which the intermolecular forces are stronger?

Is there any molecular weight dependence of the effects observed?

Version 1:

Reviewer comments:

Reviewer #1

(Remarks to the Author)

In their revised manuscript and its supplementary information, the authors responded nicely to my concerns by giving more throughout discussions and additional figures. Thus, I am now positive for the revised version to be accepted in this journal.

Reviewer #2

(Remarks to the Author)

Despite their long responses to the reviews, the authors have actually only made few changes to the manuscript, and I consider it still significantly deficient. I thought that the experimental results were interesting, but Reviewer 1 has cast doubts on those. I consider the theory to be sloppy and unconvincing. The authors still claim, in the Abstract, that they "show that these three phenomena are related", while they really don't (which they even admitted in their responses: "We agree that in our manuscript we only indicated/suggested the possibility of unifying various concepts, i.e., the concepts of nucleation, surface premelting, and finite-size melting point depression. Of course, a thorough proof of such unification requires a profound theoretical development, which is beyond the scope of this study."). In that case it is unacceptable to write "we show that these three phenomena are related". Just because two phenomena have a similar simple T-dependence doesn't show that they are related. The theory remains at a disappointingly low level, mostly just citing diverse sources (see Table 1 and the scant text below it). The title is also still inaccurate -- the authors simply refused to change it to "Temperature dependence of liquid edge layers of supported polyethylene filaments".

The authors also have not changed the misleading claim that Table 1 shows "the characteristic length scale, represented as l , of a liquid layer on the surface of a crystal, as predicted by three different phenomena", while at least two of the entries are actually the limiting thicknesses of crystals. The response claims "regarding the interpretation of characteristic length scales in Figure 5c and Table 1. We agree that the caption may have been misleading in its presentation. We have thus improved its clarity." but there is no pertinent yellow highlighted change indicated in either caption (and the caption of Table 1 does remain misleading). This seems deceptive.

On p. 2, we still read "thickness l of the wetting layer" while the Abstract described "molten regions of rather uniform width, W_{soft} " - how are l and W_{soft} different? The response says unconvincingly that one is the symbol for the experimental, the other for the theoretical thickness -- where in science do we introduce different symbols like that? It still seems that generic l is introduced to hide the fact that sometimes it refers to a crystal thickness and sometimes to the thickness of a liquid wetting layer. If in the table instead of " $l \sim$ " the authors wrote " $W_{\text{soft}} \sim$ " followed by a crystallite thickness, the inconsistency would become apparent. On p. 13, r^* is confusingly added to the mix of symbols, but it is treated as l in Table 1.

No answer was even attempted to my question regarding premelting: why is it favorable thermodynamically that at the amorphous-crystalline interface, part of the crystal is eaten away as temperature is raised, quite analogous to the observations in this manuscript? The long "response" simply goes off on tangents.

I had pointed out that the Introduction does not provide a motivation for this work. While the authors tell us that they "fully agree that providing a wide-ranging motivation at the beginning of the Introduction will significantly improve the manuscript

by framing our work within a broader scientific context. Thus, we have revised the opening paragraph to articulate the fundamental questions and the broader implications of studying crystal-melt coexistence, the critical size required for (secondary) nucleation and its relation to surface melting and the melting point reduction in nanoscopic polymer filaments.” no motivation indicating the importance of this work beyond the “ivory tower” has been added. In a polymer journal this might be acceptable, but in a general journal like Nature Communications it is not.

I had asked if an observed very thin crystal (a bright region about 7 nm across) at high temperature is compatible with the Gibbs-Thompson equation, which tells us that such thin crystals melt away, in particular at high T. The authors acknowledged the presence of this thin crystal but did not answer my question. They deflected by referring to the filament thickness, but the Gibbs-Thompson equation is about the crystal thickness. This thickness needs to be accounted for quantitatively. Otherwise the claim of equilibrium in the manuscript title and elsewhere is in serious doubt.

In summary, the manuscript remains deficient. It still requires major revisions before it can be considered for publication.

Reviewer #3

(Remarks to the Author)

My comments and queries have been adequately answered. I can recommend acceptance, although I would carefully revise the writing to make it clearer and easier to understand.

Version 2:

Reviewer comments:

Reviewer #2

(Remarks to the Author)

I continue to perceive the authors' verbose responses mostly as obfuscation. If they contain valid arguments, then those should be added to the manuscript text.

For instance, I had asked how the presence of a very thin crystal (a bright region about 7 nm across) at high temperature is compatible with the Gibbs-Thompson equation, which tells us that such thin crystals melt away, in particular at high T. The authors have given a long response and I cannot tell if they're obfuscating or giving a deep explanation that says that the Gibbs-Thompson equation for the melting point of a thin crystal does not apply when the crystal is in a filament. They don't even refer to the Gibbs-Thompson equation at all. The response ends in:

“Therefore, to specifically answer the reviewer's question: Yes, a bright region at high T is compatible as long as $2 \cdot W_{\text{soft}}(T) < w$. In other words, when $2 \cdot W_{\text{soft}}(T) \approx w$, then the whole filament is molten.” This deflection to “a bright region” rather than a thin crystal makes me suspicious that they may have been evading my question about the stability of a thin crystal at high temperature.

The authors also wrote in their response:

“Therefore, we believe a general critique of these fundamental theories is not justified. Regardless of the intended focus of the criticism, we maintain that our specific application of these theories to interpret our experimental data is both appropriate and insightful. We and the other two reviewers could not identify anything “sloppy and unconvincing” in these concepts and their theoretical descriptions.”

The first and last sentence is a quite obvious (and upsetting) misrepresentation of my evaluation. Quite clearly, my critique is not of the theories, but of their superficial presentation and sloppy and unconvincing application in this paper.

My conclusion remains that in this paper, talented experimentalists have recorded state-of-the-art images, and then someone added a superficial theoretical sauce mixing various concepts from the literature without a deeper understanding. The authors apparently lack the theoretical background to properly apply these theories, which they admitted in their earlier responses: “We agree that in our manuscript we only indicated/suggested the possibility of unifying various concepts, i.e., the concepts of nucleation, surface premelting, and finite-size melting point depression. Of course, a thorough proof of **such unification requires a profound theoretical development, which is beyond the scope of this study.**”

This limited theoretical understanding was also revealed by basic mistakes in their initial description of the quantities in their equations, which I pointed out and they conceded (which, of course, did not raise the level of the authors' theoretical understanding).

Nevertheless, the authors keep stubbornly defending their superficial conflation of various theories, including their claim, in the Abstract, that they “*show* that the three phenomena are related”, with verbose and obfuscating responses but too few substantive changes in the text. They could easily have changed the wording to say that their analysis *suggests* that the phenomena are related. The theories even have different T-dependencies, see Table 1, and “fit” only because the experimental error margins are large, see Figure 5(b). I cannot support publication of such unconvincing work of doubtful quality.

Reviewer #3

(Remarks to the Author)

I have carefully revised the rebuttal letter the authors wrote to address Reviewer 2's queries.

In my opinion, the answers are scientifically sound and quite reasonable. I would say the authors responded 90% of the concerns quite satisfactorily.

It seems Reviewer 2 is primarily concerned about some strong claims that could be toned down slightly, but I honestly think this paper deserves publication in Nature Communications.

From the Polymer Physics perspective, this work reports interesting results, and the authors have tried to explain them in the framework of existing theories. Their explanation may not be the only possible one, but it is certainly plausible.

Detailed response on comments (*written in blue*) from Reviewer #1

The authors, based on intermittent AFM imaging, reported the coexistence of crystalline phase and molten phase of PE. The sample preparation and AFM imaging were carefully performed and therefore I can find some value in this manuscript, while I found some critical error in interpreting their AFM results. Thus, I cannot recommend for the manuscript in the current form to be published in this journal.

First and foremost, we extend our sincere gratitude to Reviewer #1 for the time and effort dedicated to reviewing our manuscript, as well as for providing highly valuable, insightful and constructive feedback. We are profoundly encouraged by the positive assessment of our experiments. We fully acknowledge that the interpretation of our AFM data, particularly in the Supplementary Information, was not sufficiently elaborated. Accordingly, to address all the points raised by Reviewer #1, we have incorporated a more comprehensive discussion of these results in the revised manuscript.

Judging from the temperature-dependent change in W_{soft} , the authors caught some realistic change of those soft molten regions. However, judging from force-distance curves in Fig. SI-4, I found that the situation is not so simple.

We sincerely thank the reviewer for this insightful observation and for recognizing that the changes captured by the temperature-dependent variation in W_{soft} are realistic. We agree with the comment that the force–distance curves presented in Fig. SI-4 reveal additional complexity in the mechanical behavior of the soft molten regions. As this “extra” data was exclusively included in the Supplementary Information, we did not provide a comprehensive or detailed discussion of it in the main text. In the following section, in response to the reviewer’s comment, we address in more detail the complexity in the mechanical behavior by incorporating a thorough analysis and an extended discussion of the force–distance curves presented in Fig. SI-4.

The above figure was clipped from their Fig. SI-4(a). They wrote that the curves 1 and 2 was obtained on the mica substrate, while those are not the shape of curves on pure mica. After the jump-in contact, there is a region with almost no change in force value (solid ellipse). This type of regions always observed when surface (or AFM probe itself) is contaminated. This contamination can be the molten state PE. It also can be adsorbed water molecules on mica. Or, something else. Or, the intercalation of solvent molecules happened during spin-coating. The probe surface can also be contaminated. The similar doubt was deduced from the withdrawing curve. The long tail (dashed ellipse) will never appear for the contact between pure mica and clean AFM tip.

We sincerely thank the reviewer for the thorough and insightful comments regarding the force–distance curves shown in Fig. SI-4. We agree that the shapes of these curves deviate from those typically acquired on pure mica. We appreciate the reviewer’s careful analysis. In response, we provide the following interpretation of the observed phenomena:

As shown in Fig. R1.1-1, the "nearly-constant force region" (indicated by a solid ellipse) and the "extended tail region" (indicated by a dashed ellipse) are observed on the deflection–distance measurement on a freshly cleaved mica surface with a new cantilever having a clean AFM tip (no contamination). We attribute the apparent anomalies in the curves to the formation of a water meniscus between the AFM tip and the sample. Under ambient conditions, the water meniscus is caused by Kelvin condensation, which occurs universally. Thus, we observe water menisci also

for the PE filaments, providing a consistent explanation for the nearly-constant force region (solid ellipse) observed immediately after the jump-to-contact. Similarly, the extended tail visible during retraction (dashed ellipse) is a characteristic feature of contact with a water meniscus which eventually ruptures.

Figure R1.1-1: Deflection-distance traces taken with the same PPP-NCSTR cantilever tip. The measurement was first done on a freshly cleaved mica surface using a new AFM cantilever tip (top). Then, with the same cantilever tip, measurement was done on the region of the mica substrate surrounding our UHMWPE filaments (bottom). During both measurements, the surrounding humidity was 34% RH, and the temperature was 22 °C.

Furthermore, a comparison between deflection–distance traces obtained on freshly cleaved mica and on the region of the mica substrate surrounding our UHMWPE filaments revealed a notably narrower region of a nearly-constant force on the pure mica substrate. This difference suggests the

presence of an extremely thin, uniform layer of molten UHMWPE covering the mica surface in our samples. The uniformity of this molten UHMWPE layer is demonstrated by the homogeneous value of the AFM phase within the whole region of the mica substrate surrounding our UHMWPE filaments, which was observed consistently throughout our study. The existence of such an ultrathin layer of molten UHMWPE is expected due to the strongly different values of the surface tension of mica and polyethylene, lowering the surface tension of the region of the mica substrate surrounding our UHMWPE filaments. Additionally, a pronounced jump in the deflection of the cantilever (marked by the red arrows in Fig. R1.1-1) is evident in both approach and retraction traces on fresh mica, but appears only in the retraction direction of the curves measured on the region of the mica substrate surrounding our UHMWPE filaments. This discrepancy occurs because, as the tip approached the surface, the ultrathin layer of molten UHMWPE initially “shields” the tip from strong attractive interactions with the actual (bare) mica surface. However, as this layer was penetrated during further extension, upon subsequent retraction the trace exhibited a sudden detachment event when the tip lost contact with this layer. Moreover, due to the ultrathin layer of molten UHMWPE, this detachment “jump” during retraction is less sharp on the UHMWPE-covered mica than on freshly cleaved mica.

We appreciate the reviewer’s concern regarding the potential influence of residual solvent molecules on the sample surface. We would like to clarify that, although the presence of trace residual solvent cannot be ruled out entirely, any such remaining amount is expected to be small and probably negligible. The spin-coating process employed para-xylene, a highly volatile solvent that evaporates readily under ambient conditions. Furthermore, many of the samples have been stored unsealed in ambient atmosphere for over six months before performing measurements, with constant air exchange ensured by continuous ventilation. Under these conditions, we are confident that residual solvent effects, if any, do not affect the interpretation of our results.

Moreover, the homogeneous and uniform signal observed in the AFM phase images of the whole region of the mica substrate surrounding our UHMWPE filaments supports the robustness of our data. Even if some level of cantilever tip contamination were present, its influence would manifest as a systematic offset in the phase values rather than as random or selective artifacts. Therefore, while we acknowledge the possibility of the cantilever tip getting contaminated by the UHMWPE melt, such contributions would not alter the central conclusions drawn from our experimental

observations. We appreciate the reviewer's astute observations, which have helped us clarify these important mechanistic details.

The following information has been added at the end of section 2 of the Supplementary Information.

Fig. SI-5: Deflection-distance traces taken with the same PPP-NCSTR cantilever tip. The measurement was first done on a freshly cleaved mica surface using a new AFM cantilever tip (top). Then, with the same cantilever tip, measurement was done on the region of the mica substrate surrounding our UHMWPE filaments (bottom). During both measurements, the surrounding humidity was 34% RH, and the temperature was 22 °C.

In force spectroscopy measurements conducted on the region of the mica substrate surrounding our UHMWPE filaments — specifically, lines 1, 2, and 10 in Fig. SI-4(a) — a distinctive feature was observed: an area with nearly constant force value following the jump-in to contact (indicated by a solid ellipse) accompanied by a long tail in the retraction curve (indicated by a dashed ellipse).

This suggests the presence of a liquid medium between the sample surface and the AFM tip. Possible sources for this liquid medium include a water meniscus formed via Kelvin condensation, a thin layer of molten polyethylene (PE), or contamination of the cantilever tip by molten PE chains.

To identify the nature of this liquid medium, deflection–distance measurements were performed using a new cantilever having a clean AFM tip (no contamination) on a freshly cleaved mica surface. The same cantilever tip was then used to perform measurements on the mica region of the UHMWPE sample. As shown in **Fig. SI-5**, the "nearly-constant force region" (indicated by a solid ellipse) and the "extended tail region" (indicated by a dashed ellipse) were observed on the deflection–distance measurement on a freshly cleaved mica surface with a new cantilever having a clean AFM tip (no contamination). We attribute the apparent anomalies in the curves to the formation of a water meniscus between the AFM tip and the sample. Under ambient conditions, the water meniscus is caused by Kelvin condensation, which occurs universally. Thus, we observed water menisci also for the PE filaments, providing a consistent explanation for the nearly-constant force region (solid ellipse) observed immediately after the jump-to-contact. Similarly, the extended tail visible during retraction (dashed ellipse) is a characteristic feature of contact with a water meniscus which eventually ruptures.

Furthermore, a comparison between deflection–distance traces obtained on freshly cleaved mica and on the region of the mica substrate surrounding our UHMWPE filaments revealed a notably narrower region of a nearly-constant force on the pure mica substrate. This difference suggests the presence of an extremely thin, uniform layer of molten UHMWPE covering the mica surface in our samples. The uniformity of this molten UHMWPE layer is demonstrated by the homogeneous value of the AFM phase within the whole region of the mica substrate surrounding our UHMWPE filaments, which was observed consistently throughout our study. The existence of such an ultrathin layer of molten UHMWPE is expected due to the strongly different values of the surface tension of mica and polyethylene, lowering the surface tension of the region of the mica substrate surrounding our UHMWPE filaments. Additionally, a pronounced jump in the deflection of the cantilever (marked by the red arrows in **Fig. SI-5**) is evident in both approach and retraction traces on fresh mica, but appears only in the retraction direction of the curves measured on the region of the mica substrate surrounding our UHMWPE filaments. This discrepancy occurs because, as the AFM tip approached the surface, the ultrathin layer of molten UHMWPE initially “shields” the tip

from strong attractive interactions with the actual (bare) mica surface. However, as this layer was penetrated during further extension, upon subsequent retraction the trace exhibited a sudden detachment event when the tip lost contact with this layer. Moreover, due to the ultrathin layer of molten UHMWPE, this detachment "jump" during retraction is less sharp on the UHMWPE-covered mica than on freshly cleaved mica.

If so, what is phase zero in their intermittent AFM? The baseline is not any more the real baseline.

We thank the reviewer for raising this important question regarding the baseline in intermittent contact AFM measurements. We appreciate the opportunity to present further details, which help to clarify our approach.

In our analysis, as the absolute phase baseline may vary between scan lines due to the exact absolute amplitude of the cantilever oscillation influenced by thermal drift or environmental noise, the value of the phase signal representing the mica substrate surrounding our UHMWPE filaments was used as a reference, specifically for flattening the AFM phase images. It is also important to note that taking the region of the mica substrate surrounding our UHMWPE filaments as the reference surface, this region may include additional interfacial layers, such as a thin layer of molten UHMWPE melt, contributions from the water meniscus formed under ambient conditions, or even potential tip contamination from the UHMWPE melt. Nevertheless, as we always used the same referencing procedure under these conditions, our interpretation remains both methodologically consistent and physically justified, in particular as the overall response of the phase signal of this composite reference system remains stable and invariant over time. Crucially, any systematic shift in the reference value does not affect our interpretation of the presented phase contrasts relative to this reference value. The clear distinction in phase values between the filament and the mica substrate surrounding our UHMWPE filaments, as well as the contrast between the two distinct regions within the filament itself, remains unequivocally discernible regardless of baseline adjustments.

The following information has been added at the end of section 2 of the Supplementary Information.

Knowing that both, a water meniscus and an ultra-thin polyethylene (PE) melt layer cover the entire UHMWPE sample, a critical question arises: whether the phase value from the mica substrate surrounding our UHMWPE filaments can still serve as a reliable reference for flattening the AFM phase data, and, if this flattening is necessary?

As the absolute phase baseline may vary between scan lines due to factors such as thermal drift or environmental fluctuations, the value of the phase signal representing the mica substrate surrounding our UHMWPE filaments was used as a reference, specifically for flattening the AFM phase images. It is also important to note that taking the region of the mica substrate surrounding our UHMWPE filaments as the reference surface, this region may include additional interfacial layers, such as a thin layer of molten UHMWPE melt, contributions from the water meniscus formed under ambient conditions, or even potential tip contamination from the UHMWPE melt. Nevertheless, as we always used the same referencing procedure under these conditions, our interpretation remains both methodologically consistent and physically justified, in particular as the overall response of the phase signal of this composite reference system remains stable and invariant over time. Crucially, any systematic shift in the reference value does not affect our interpretation of the presented phase contrasts relative to this reference value. The clear distinction in phase values between the filament and the mica substrate surrounding our UHMWPE filaments, as well as the contrast between the two distinct regions within the filament itself, remains unequivocally discernible regardless of baseline adjustments.

Furthermore, these trends are also observed event at the top of “hard” filaments. How they interpret it? Every discussion made in this manuscript need to be modified.

We sincerely thank the reviewer for this insightful observation and for requesting further clarification. Similar to the other regions of the examined filaments, the presence of the "nearly-constant force region" and the "extended tail region" atop the "hard" filaments can be attributed to several factors: the formation of a water meniscus between the sample surface and the cantilever tip under ambient conditions, possible tip contamination from the UHMWPE melt, or the presence

of a thin layer of UHMWPE melt. As we discuss later in this reply in the context of “non-melting crystalline surfaces”, we anticipate that the top of the hard filaments is either a non-melting surface which is not covered by any liquid layer or covered with an extremely thin layer of UHMWPE melt which, under the conditions of our AFM measurements, is easily penetrated by the AFM tip.

Although the figure is not in the main text and therefore this is just a comment, but the vertical axis is somewhat strange. When I converted the force value into deflection value for curve 10 (assume that the spring constant is 5 N/m), the deflection difference between D of 5 nm and 10 nm is more than 5 nm. Thus, the sensitivity calibration was wrongly performed.

We appreciate the reviewer’s careful attention to the calibration details. In response to the comment regarding the vertical axis and inverse optical sensitivity calibration, we would like to clarify that our calibration procedure yields values that are physically reasonable. For instance, taking the same curve 10 as an example and assuming a spring constant of 5 N/m, the corresponding force change measured over 5 nm displacement (in Figure R1.1-2, this corresponds to the range from 10 nm to 5 nm in distance) yields approximately 25 nN, as illustrated in Figure R1.1-2. This result aligns well with theoretical expectations and supports the validity of our calibration approach.

Figure R1.1-2: For the calibration of the spring constant of the cantilever

Second, I love their discussion using Fig. 2. Very careful. However, if looking at Figs. SI-4 and SI-7, soft layers are not symmetric especially when the scan direction is perpendicular to the direction of the filament. This is surely due to the tip-shape effect. I know this portion is not so easy, but the authors should mention something about the effect. If possible, they may be able to deconvolute the tip effect and once again discuss the value of W_{soft} .

We sincerely thank the reviewer for the highly positive feedback regarding the discussion of Fig. 2, as well as for raising the important point concerning the asymmetry in the width observed in the soft layers in Supplementary Figs. SI-4 and SI-7, particularly when scanning perpendicular to the filament axis. We agree that this asymmetry is likely influenced by tip-shape effects and scanning orientation as well as scanning direction relative to the height gradient of the sample.

In response to this comment, in Figure R1.1-3, we have included both trace and retrace phase images for two representative filaments. The observed asymmetries are mainly related to the direction of scanning over the sample topography. Specifically, when scanning perpendicular to the filament, the tip encounters an ascending and then descending slope, resulting in phase profiles in the trace and retrace directions that exhibit “mirror-image asymmetry” — as clearly demonstrated in Figure R1.1-3.

To minimize potential errors introduced by this effect, all values of W_{soft} reported in the manuscript were measured from cross-sectional phase profiles obtained when the cantilever was scanning in the direction of **ascending** the soft layer slope. For instance, as illustrated in Figure R1.1-3, the width of the left soft layer was measured from the trace scan (where the tip ascends the left slope), while the right soft layer was measured from the retrace scan (where the tip ascends the right slope). This approach ensured consistent and reliable quantification of the soft layer width, reducing artifacts related to scan direction.

We acknowledge that tip deconvolution represents a valuable future direction for refining our analysis. We thank the reviewer for highlighting this possibility for further methodological development. However, for the central message of our manuscript concerning the change in width of the soft layer with temperature, such a deconvolution does not seem to be required.

Figure R1.1-3: AFM phase trace and corresponding retrace images indicate that the observed asymmetry is a result of whether the AFM tip is scanning in trace or retrace direction. Trace (a) and retrace (c) AFM phase images taken from the same filament (taken in a single AFM measurement under ambient conditions), similar to the conditions used for obtaining results shown in Fig. SI-7. Both images have the same color code and scale bar. Trace (b) and retrace (d) AFM phase images taken from the same filament (taken in one AFM measurement under ambient conditions), similar to the condition used for obtaining results shown in Fig. SI-4. Both images have the same color code and scale bar. The scan direction used for obtaining these images is labeled on their top (bottom) with a red arrow.

A new section 8 (see below) has been added to the Supplementary Information.

8. Dependence of the AFM phase signal on the scanning orientation relative to the height gradient of the sample.

Fig. SI-11: AFM phase trace and corresponding retrace images indicate that the observed asymmetry is a result of whether the AFM tip is scanning in trace or retrace direction. Trace (a) and retrace (c) AFM phase images taken from the same filament (taken in a single AFM measurement under ambient conditions), similar to the condition of Fig. SI-8. Both images have the same color code and scale bar. Trace (b) and retrace (d) AFM phase images taken from the same filament (taken in one AFM measurement under ambient conditions), similar to the condition of Fig. SI-4. Both images have the same color code and scale bar. The scan direction of those images is labeled on their top (bottom) with a red arrow.

The asymmetry observed in the soft layers in Supplementary Figs. SI-4 and SI-8, particularly when scanning perpendicular to the filament axis, are likely influenced by the scanning orientation relative to the height gradient of the sample.

In **Fig. SI-11**, we have included both trace and retrace phase images for two representative filaments. The observed asymmetries are mainly related to the direction of scanning over the sample topography. Specifically, when scanning perpendicular to the filament, the tip encounters an ascending and then descending slope, resulting in phase profiles in the trace and retrace directions that exhibit “mirror-image asymmetry” — as clearly demonstrated in **Fig. SI-11**.

To minimize potential errors introduced by this effect, all values of W_{soft} reported in the manuscript were measured from cross-sectional phase profiles obtained when the cantilever was scanning in the direction of **ascending** the soft layer slope. For instance, as illustrated in **Fig. SI-11**, the width of the left soft layer was measured from the trace scan (where the tip ascends the left slope), while the right soft layer was measured from the retrace scan (where the tip ascends the right slope). This approach ensured consistent and reliable quantification of the soft layer width, reducing artifacts related to scan direction.

Finally, this comment is not so critical but they should clearly describe the resonant frequency of the cantilever and actual vibration frequency. They may better to describe how to measure the resonant frequency. Just very far away from the sample surface like written in the manual of AFM or perform it at the vicinity of surface? I am asking this question because they used very small amplitude ratio. The resonant frequency changes a lot due to long-range interaction force or air damping. If they were not so careful, all the measurement is not in “soft tapping” region but rather strong force is exerted on the specimen.

We thank the reviewer for raising this pertinent point regarding the characterization of the cantilever’s resonance frequency and vibration parameters, which are essential to ensure that measurements were conducted within the soft tapping regime. We fully agree that careful calibration is crucial to minimize the influence of long-range forces and environmental factors.

In our experiments, the resonance frequency of the cantilever (approximately 165 kHz) was initially determined at a minimum distance of 100 μm away from the sample surface to avoid any (also long range) tip–sample interactions. To account for initial instrumental instabilities like thermal drifts, all data collected within the first hour of operation were excluded from our study. Immediately before acquiring the data presented in the manuscript and in the Supplementary

Information, we re-measured both the resonance frequency and vibration amplitude again under the same non-interactive conditions (at a distance $\geq 100 \mu\text{m}$ above the surface). Furthermore, the free vibration amplitude was verified after each measurement using the procedure detailed in Fig. SI-9(a), where the plateau region of the amplitude-distance curve provides a direct measure of this parameter. This systematic approach ensured that the operating parameters remained within the soft tapping regime throughout the measurements, thereby minimizing the force exerted on the sample.

In the "Methods- Characterization techniques" section of our manuscript, “a characteristic spring constant of the cantilever of around 48 N/m, and a resonance frequency of the cantilever of around 165k Hz” has been added to the first paragraph.

In the "Methods- Characterization techniques" section of our manuscript, “obtained at a position where the cantilever tip was at least 100 μm from the sample surface” has been added to the third paragraph, the information “the AFM cantilever oscillates at a frequency slightly lower (in the order of 100 Hz) than one of the resonance frequencies” has been added. The full sentence now reads as: “During a typical intermittent-contact mode AFM measurement (in air), the AFM cantilever oscillates at a frequency slightly lower (in the order of 100 Hz) than one of the resonance frequencies with a "free" amplitude A_0 , obtained at a position where the cantilever tip was at least 100 μm from the sample surface.”

Detailed response on comments (*written in blue*) from Reviewer #2

This paper provides convincing evidence for ~9-nm thin liquid layers along the edges of mica-supported semicylindrical polyethylene filaments, obtained by AFM, and tries to explain their temperature dependence. This is an intriguing system, which was apparently introduced earlier (see ref. 57 from 2015).

We are deeply grateful to the reviewer for the overall positive assessment of our work. We truly appreciate the recognition of the convincing evidence for the ~9-nm width liquid layers along the edges of the mica-supported semi-cylindrical polyethylene filaments obtained by AFM, as well as the interest in our explanation of the temperature dependence of these structures. It is particularly encouraging to us that the reviewer finds our studied system intriguing, and we are thankful for acknowledging its connection to some of our earlier studies. In ref. 57, polymer filaments of several micrometers were generated by a dewetting process. In the present manuscript, we used spin coating under different conditions, allowing that polymer chains undergo a coil-to-stretch transition and forming ultra-long filaments of up to millimeters. We would like to emphasize that we anticipate that most polymer chains were (almost) fully stretched, reducing the number of possible chain folds and chain entanglements within the filaments drastically. Thus, the thin liquid layers along the edges of mica-supported semi-cylindrical polyethylene filaments are expected to be in contact with a crystalline surface. This is in contrast to a possible liquid-like layer on the fold surface of a lamellar polymer crystal and is also different from a so-called “rigid-amorphous fraction”, mainly located at the fold surface of polymer crystals.

The title “Temperature dependence of crystal – melt coexistence for supported polyethylene filaments” does not describe the content of the paper accurately. (Below I’m proposing some experiments that could help fulfill the promise of this more ambitious title.) As it stands, “Temperature dependence of liquid edge layers of supported polyethylene filaments” would be an accurate title.

We sincerely appreciate the reviewer’s thoughtful suggestion regarding the title of our manuscript. We agree that clarity and accuracy are essential. The proposed alternative title, “Temperature Dependence of Liquid Edge Layers of Supported Polyethylene Filaments,” certainly captures our

key experimental observation. At the same time, we believe that the original title, “Temperature Dependence of Crystal–Melt Coexistence for Supported Polyethylene Filaments,” is better suited because it captures in addition the essential physical phenomenon underlying the formation of the liquid edge layers—namely, the **equilibrium** coexistence between crystalline and molten phases characterized by a length scale which depends on temperature.

The theory presented centrally in this paper requires major revisions. Already in the Abstract, the authors name-drop “nucleation, surface premelting, and the melting-point depression of finite-size”, which are later confirmed to refer to the critical nucleus in classical nucleation theory (a lower limit of crystal size), premelting, and Gibbs-Thompson theory (another more stringent lower limit of crystal size). While at the end of Abstract and elsewhere, the authors claim to have unified these phenomena, the theory remains at a disappointingly low level, mostly just citing diverse sources (see Table 1 and the scant text below it).

We sincerely appreciate the reviewer’s critical comment and the opportunity to clarify the scope and intention of our discussion of related theoretical concepts. We thank the reviewer for the careful reading and acknowledge that our brief and limited presentation of the theoretical context may have caused misunderstandings.

We agree that in our manuscript we only indicated/suggested the possibility of unifying various concepts, i.e., the concepts of nucleation, surface premelting, and finite-size melting point depression. Of course, a thorough proof of such unification requires a profound theoretical development, which is beyond the scope of this study. The primary goal of our work presented in the manuscript was to provide a possible interpretation of the observed crystal–melt coexistence in polyethylene filaments. Thus, we have compared our results with predictions of the Lauritzen–Hoffman theory of secondary nucleation at the growth front of the crystal. References to classical nucleation theory, surface premelting, and the Gibbs–Thomson effect were included to indicate that these phenomena are related and can, within the uncertainty of our derived results on the widths of the soft region, also explain our observations.

CNT and Gibbs-Thompson theories for crystal size are not really applicable to the thickness of a liquid layer studied here.

We thank the reviewer for raising this important point regarding the applicability of classical nucleation theory (CNT) and the Gibbs–Thomson equation to interpret the thickness of the liquid layer along the edges of mica-supported semi-cylindrical polyethylene filaments observed in our study. This comment allows us to provide more details on the theoretical links between a stringent lower limit of crystal size and the equilibrium thickness of the liquid layer (or, e.g., in the context of ice and water, also called quasi-liquid layer) on the surface of a crystal, as discussed and predicted in concepts of surface melting or pre-melting. As described in a large number of publications and also demonstrated in recent computer simulations on nanoscopic systems, in equilibrium, a crystal can be covered by a liquid layer of its melt. As shown, for example, in ref (1) of our manuscript, U. Tartaglino, T. Zykova-Timan, F. Ercolessi, E. Tosatti, **Melting and nonmelting of solid surfaces and nanosystems**, Phys. Rep. 411 (2005) 291–321, <https://doi.org/10.1016/j.physrep.2005.01.004> , the thickness $\ell(T)$ of this liquid layer can be related to the wetting behavior described via the corresponding surface and interfacial tensions. For curved interfaces, one has to include additional terms like, for example, a term characterized by the Tolman length (see, e.g., P. Montero de Hijes, Jorge. R. Espinosa, Valentino Bianco, Eduardo Sanz, and Carlos Vega, **Interfacial Free Energy and Tolman Length of Curved Liquid-Solid Interfaces from Equilibrium Studies**, J. Phys. Chem. C, 124, 8795-8805 (2020); <https://doi.org/10.1021/acs.jpcc.0c00816>). Represented through an “interaction” free energy term $V(\ell)$, which includes the various short- and long-range intermolecular forces at work, the solid–liquid and the liquid–vapor interfaces “feel” their mutual presence. As described through equation (2) in this above-mentioned paper by Tartaglino et al., the thickness $\ell(T)$ of this layer diverges as the melting temperature is approached. Thus, a crystal of finite size σ will be molten completely when the molten layer covers the whole size of the crystal., i.e., $\ell(T) \geq \sigma$. Consequently, similar to what we showed in Fig. 5 of our manuscript, the value of $\ell(T)$ decides also the melting temperature of finite size crystals.

We agree that it may not be obvious how nucleation and surface melting are related. The underlying assumption is that nucleation is not possible when the equilibrium size of the coexisting liquid phase in a finite system is equal or even larger than the size σ of the whole system. As shown

in refs. (14) - (16) cited in our manuscript (and, for convenience, at the end of this paragraph), for a fixed small number N of molecules and a finite volume V (i.e., a nanoscopic system) at a temperature T , it is possible to observe stable interfaces between a crystalline and a liquid phase that are in thermodynamic equilibrium. In these works, it has also been shown that this equilibrium size of the crystalline phase in **finite** systems can be related to the size of a critical nucleus in **large** (infinite) systems, where crystal growth can be observed. The temperature-dependence of the size of the critical nucleus in infinite systems is used for the comparison of our experimental results on the width W_{soft} of the liquid layers along the edges of mica-supported semi-cylindrical polyethylene filament of finite size.

(14) Binder, K.; Block, B. J.; Virnau, P.; Tröster, A. **Beyond the Van Der Waals Loop: What Can Be Learned from Simulating Lennard-Jones Fluids inside the Region of Phase Coexistence.** American Journal of Physics 2012, 80 (12), 1099–1109. <https://doi.org/10.1119/1.4754020>.

(15): Statt, A.; Virnau, P.; Binder, K. **Finite-Size Effects on Liquid-Solid Phase Coexistence and the Estimation of Crystal Nucleation Barriers.** Phys. Rev. Lett. 2015, 114 (2), 026101. <https://doi.org/10.1103/PhysRevLett.114.026101>.

(16) Montero de Hijes, P.; Vega, C. **On the Thermodynamics of Curved Interfaces and the Nucleation of Hard Spheres in a Finite System.** The Journal of Chemical Physics 2022, 156 (1), 014505. <https://doi.org/10.1063/5.0072175>.

Specifically, in the central Figure 5c, the authors use the two limiting crystal thicknesses to fit the T -dependence of a liquid layer. In the caption of Table 1, they make the misleading claim that the table shows “the characteristic length scale, represented as L , of a liquid layer on the surface of a crystal, as predicted by three different phenomena”, while at least two of the entries are actually the limiting thicknesses of crystals.

We thank the reviewer for this insightful comment regarding the interpretation of characteristic length scales in Figure 5c and Table 1. We agree that the caption may have been misleading in its presentation. We have thus improved its clarity.

As mentioned in response to the previous comment with respect to the equilibrium coexistence of a crystalline and a liquid phase in a **finite** system, the underlying assumption is that nucleation is only possible when the equilibrium size of the coexisting liquid phase in a finite system is less than the whole system. Thus, if the volume fraction of the liquid phase increases, the volume fraction of the solid phase decreases until it disappears at the melting temperature of this finite size system. Of course, geometry of the system of finite size matters but also the geometry of the (often faceted) solid phase is relevant. Thus, it can only be an approximation when we represent the respective volumes by a single “characteristic” length scale. For most theoretical concepts we used a thickness $\ell(T)$ while for our experimental results we represent this length scale by the width W_{soft} of the liquid layers along the edges of mica-supported semi-cylindrical polyethylene filament of finite size. When describing the volume fraction of the solid phase, we mainly assume that it has a cylindrical shape, allowing a description by its radius r (or r^* when we consider a critical nucleus). As shown, for example, in refs. (14) - (16) cited in our manuscript, in most cases the geometry and the curvature of the system or that of the solid phase can be accounted for by an appropriately derived prefactor. Thus, describing the phases through a single length scale is a useful approximation, similar to the concept of the Tolman length scale which was introduced to account for the impact of curved interfaces on nucleation via changes in interfacial energies quantified through a length scale, i.e., the Tolman length.

In our manuscript, we do not intend to merge distinctly different concepts. However, we want to illustrate that the same (mean) intermolecular forces acting at interfaces, which vary with the density of solid and liquid phases, are responsible for different phenomena, determining characteristic features of the coexistence of a crystal with its melt. Thus, representing this coexistence by a characteristic length scale allows us to compare, at least qualitatively, the predicted **temperature dependence** of the various concepts.

On p. 2, we read “thickness l of the wetting layer” while the Abstract described “molten regions of rather uniform width, W_{soft} ” - how are l and W_{soft} different? It seems that generic l is introduced to hide the fact that sometimes it refers to a crystal thickness and sometimes to the thickness of a liquid wetting layer. If in the table instead of “ $l \sim$ ” the authors wrote “ $W_{\text{soft}} \sim$ ” followed by a crystallite thickness, the inconsistency would become apparent. On p. 13, r^ is confusingly added to the mix of symbols, but it is treated as l in Table 1.*

We sincerely thank the reviewer for this thoughtful question, which helps us clarify an important distinction in our terminology. The parameter " ℓ " in our manuscript refers to the theoretically predicted thickness of a molten polymer wetting layer on an idealized, smooth crystalline substrate, as derived from various thermodynamic models. In contrast, " W_{soft} " represents the experimentally measured width of the soft, molten PE meniscus layer formed between the PE nanofilament and the mica substrate, derived from our AFM phase data. In our answers to the following comments, we will clarify further the meaning and relations between the various length scales introduced and discussed in our manuscript.

The theory on p. 13 contains technical flaws in the definition of the quantities (see details below), which suggests that the authors were struggling with the concepts. It seems that the simple theory on p. 13 is suitable for primary nucleation, but the authors make it clear that they want to consider secondary nucleation. In secondary nucleation, there should be two characteristic length scales, along the preexisting surface and perpendicular to it, not just a single r^ . I see this confirmed in the corresponding author's cartoon of a secondary critical nucleus in reference 50, which is one chain stem thick and five stems wide. The assumption of a cylindrical secondary nucleus (top of p. 12) is unreasonable.*

We sincerely thank the reviewer for this insightful comment and for providing the opportunity to clarify our theoretical approach. We appreciate the careful reading and acknowledge the need for more detailed descriptions for the CNT.

In the initial version of the manuscript, the critical size of the secondary nucleus was modeled as a cylinder with volume $V_{\text{cr}} = \pi r^{*2} \cdot l^*$. There, two characteristic length scales are used: " r^* " is the radius and " l^* " is the length of the cylinder. Given that the polymer chains within the nanofilament are oriented along its axis, and considering that the orientation of secondary nuclei is governed by the crystalline face of original crystal—which is aligned along the nanofilament—we consequently assume that the polymer chains inside the secondary nuclei are likewise aligned parallel to their respective nanofilaments. As a result, " l^* " is oriented parallel to the nanofilament, while " r^* " is perpendicular to it. It is therefore " r^* " that governs the value of " W_{soft} ", and for this reason, our subsequent analysis focuses specifically on this parameter.

We fully agree that shape of the secondary nucleus for polymers is not simple and may involve multiple length scales. In reference [50] and also in I Liu, Y., Wang, Z., Zhang, Y. et al. **The size of critical secondary nuclei of polymer crystals does not depend on supersaturation**. Nat Commun 16, 3773 (2025). <https://doi.org/10.1038/s41467-025-58962-5>, we have given some examples. We appreciate the reviewer's emphasis on this point. Accordingly, we revised the relevant section of the manuscript to clearly justify the use of a cylindrical representation, explicitly define both dimensions, and explain the physical reasoning why we have focused only on " r^* " in the context of our experimental system.

The sentence "Moreover, since the secondary nuclei adopt the same orientation as the original crystal—that is, the nanofilament in which the polymer chains are aligned along its axis—the cylindrical nucleus is aligned lengthwise along the nanofilament, while its radius is oriented perpendicularly; this radius is directly correlated with the value of W_{soft} , i.e., $W_{\text{soft}} \sim r^*$. This implies that the crystal-melt coexistence observed at the growth front of the polymer crystal differs from the well-studied coexistence at the fold surface." has been added to page 12, after the sentence: "Accordingly, if the residual number of molten polymer chains at the growth front per appropriate length along the filament adds up a total volume less than V_{cr} , these polymer chains cannot crystallize and will stay in the molten state."

And why can't the existing fibrillar crystal grow without nucleation, as in Regime I of Lauritzen-Hoffman theory?

We sincerely thank the reviewer for raising this fundamental question regarding the applicability of the Lauritzen-Hoffman theory to our system. We appreciate the opportunity to clarify the role of secondary nucleation in the growth of existing nanofilament crystals.

It is our understanding that even within Regime I of the Lauritzen-Hoffman theory, crystal growth proceeds via a secondary nucleation process, albeit at a relatively slow rate compared to the subsequent substrate layer completion step. Like in many previous publications by multiple authors, the concept of secondary nucleation and its relation to homogeneous (primary) nucleation has been discussed also in a recent publication: Liu, Y., Wang, Z., Zhang, Y. et al. **The size of critical secondary nuclei of polymer crystals does not depend on supersaturation**. Nat

Commun 16, 3773 (2025). <https://doi.org/10.1038/s41467-025-58962-5> . The key difference between Regime I and Regime II lies in the rate of secondary nucleation; in Regime I, the spreading and completion of the crystalline layer are much faster than the formation of a new secondary nucleus on the growth face, leading to nearly homogeneous growth in the lateral direction. However, such homogeneous growth does not imply the absence of secondary nucleation, but rather reflects a specific kinetic relationship between its rate and the rate of surface spreading of the crystalline layer.

We have based our interpretation on the foundational work by Hoffman, Davis, and Lauritzen (as detailed in Section 3.4.2 “**Growth Rate for the Case Where the Formation of a Surface Nucleus Is Followed by Rapid Completion of the Substrate (Regime I)**” of their chapter in **Treatise on Solid State Chemistry**, Springer, 1976), which establishes that secondary nucleation is an integral component of the growth mechanism in both regimes for chain-folded polymer crystals.

The formation of fibrillar crystals follows the same concepts of nucleation. As discussed in our previous publication *, even stretched polymer chains, which are attracted via van der Waals interactions, need to form “bundles” of a certain (critical) size in order to become crystalline.

* ref. (28) in our manuscript:

Da Huang, Thorsten Hugel, and Günter Reiter, **Mechanism for the Formation of Millimeter-Long Solid Filaments by Spin Coating Dilute Solutions of a Crystallizable Polymer**, *Macromolecules* 2024 57 (8), 3696-3705; <https://doi.org/10.1021/acs.macromol.3c02524>

What the data do seem to show is premelting. It is included in Table 1 but not really discussed. It needs to be explained that while the crystal is the lowest free-energy form of the polymer, surface energy of the air-polymer interface makes insertion of a liquid-like layer thermodynamically favorable. It should also be discussed that in bulk polymers, premelting has been invoked, e.g. by Tanabe, Strobl, and Fischer ([https://doi.org/10.1016/0032-3861\(86\)90001-7](https://doi.org/10.1016/0032-3861(86)90001-7)) in studies of HDPE, with a different meaning: without air, at the amorphous-crystalline interface, eating away part of the crystal as temperature is raised, quite analogous to the observations in this manuscript – why is this favorable thermodynamically?

We would like to clarify here that the main idea of this manuscript is to understand the observed crystal-melt coexistence of supported PE nanofilaments. As a first interpretation of our experimental observations described through $W_{\text{soft}}(T)$, we have chosen CNT, employed within the concept of secondary nucleation. There, the underlying assumption is that nucleation is only possible if enough molten molecules are available to form a nucleus of critical size, approximately described through the parameter r^* . This assumption leads to the condition that no nuclei are formed inside the soft meniscus layer of molten PE, which is characterized by $W_{\text{soft}}(T)$, when $W_{\text{soft}}(T) \sim r^*(T)$, due to the limitation of the size of the meniscus, i.e., the lack of a sufficient number of molten molecules. As predicted by theory on surface melting or premelting and confirmed by various experimental systems, a crystalline surface may be covered by a liquid layer of molten molecules of the same kind. We related the characteristic length scale of this predicted liquid layer with our experimental results on the width W_{soft} of the liquid layers along the edges of mica-supported semi-cylindrical polyethylene filament of finite size (width). There, the underlying assumption is that a filament where all polymer chains would in a crystalline state at a temperature of $T = 0$ K has to be covered by molten polymer chains at $T \gg 0$ K. The relation of $W_{\text{soft}}(T)$ with the reduced melting temperature of crystals of finite size σ invokes that no crystalline phase is remaining if the width of the liquid layer of molten molecules covers the whole finite system.

As discussed in the above section, polymer chains within the nanofilament are almost exclusively oriented parallel to the filament axis. There is no surface covered with chain folds as it is exhibited in lamellar crystals of chain-folded polymers. Consequently, in the case of the investigated filaments, molten polymer chains are coexisting on the lateral surface of the crystalline core of the filaments, which may be identified as the "growth front" of the "oriented" polyethylene crystal, rather than the molten phase on the fold surface of lamellar polymer crystals, as reported by numerous other researchers, e.g., also by Tanabe, Strobl, and Fischer, **Surface melting in melt-crystallized linear polyethylene**, *Polymer*, 27, 1147-1153 (1986); [https://doi.org/10.1016/0032-3861\(86\)90001-7](https://doi.org/10.1016/0032-3861(86)90001-7).

We note that all concepts discussed above are based on thermodynamic parameters which apply to equilibrium situations. None of these concepts include any kinetic (time-dependent) aspects. Our results on $W_{\text{soft}}(T)$ show no dependence on the age of the sample, do not depend on filament

size or morphology, and have been shown to be independent of molecular weight and dispersity of the polyethylene used. The interpretation of the results on surface melting by Tanabe et al. and later versions by M. Muthukumar[†] or by J.U. Sommer[‡] also rely on equilibrium, i.e., thermodynamic concepts.

[†] Muthukumar, M. (2007). **Shifting Paradigms in Polymer Crystallization**. In: Reiter, G., Strobl, G.R. (eds) **Progress in Understanding of Polymer Crystallization**. Lecture Notes in Physics, vol 714. Springer, Berlin, Heidelberg. https://doi.org/10.1007/3-540-47307-6_1

[‡] Sommer, J.U. (2007). **Theoretical Aspects of the Equilibrium State of Chain Crystals**. In: Reiter, G., Strobl, G.R. (eds) **Progress in Understanding of Polymer Crystallization**. Lecture Notes in Physics, vol 714. Springer, Berlin, Heidelberg. https://doi.org/10.1007/3-540-47307-6_2

The literature on conventional premelting tells us that any crystal, in particular near its melting point, is covered with a liquid-like layer. Why is there no such liquid layer on top of the crystalline filament in Figure 3?

We sincerely thank the reviewer for raising this insightful question regarding the apparent absence of a liquid-like layer on top of the crystalline filament in Figure 3, which helps us clarify an important aspect of our experimental observations.

As described in theoretical treatments (see, e.g., ref (1) of our manuscript, U. Tartaglino, T. Zykova-Timan, F. Ercolessi, E. Tosatti, **Melting and nonmelting of solid surfaces and nanosystems**, Phys. Rep. 411 (2005) 291–321, <https://doi.org/10.1016/j.physrep.2005.01.004>), and demonstrated in experiments (see, e.g., ref (4) of our manuscript, Pluis, B.; van der Gon, A. W. D.; Frenken, J. W. M.; van der Veen, J. F. **Crystal-Face Dependence of Surface Melting**. Phys. Rev. Lett., 59, 2678 (1987). <https://doi.org/10.1103/PhysRevLett.59.2678>), depending on the interfacial properties, a thin molten layer could indeed form on the top surface of the crystalline nanofilament. However, besides surface melting, there exist also conditions for non-melting of solid surfaces. Even within a single crystal (see, for example, lead), some crystal faces may exhibit surface melting but other faces are not covered by a liquid layer (see, for example, D. Schebarchov and S. C. Hendy, **Superheating and Solid-Liquid Phase Coexistence in Nanoparticles with Nonmelting Surfaces**, Phys. Rev. Lett. 96, 256101 (2006); <https://doi.org/10.1103/PhysRevLett.96.256101>), leading to the curious observation of superheating of such crystalline faces.

Due to the semi-cylindrical geometry of the nanofilament and its interaction with the wettable mica substrate, we have to deal with different values of the interfacial energies and surface tensions across the filament. Thus, we may anticipate that not all parts of the filament are covered with a liquid layer and the conditions for surface melting (or non-melting) are not fulfilled for all parts of the nanofilament. Thus, probably due to the opposite curvature at the top of the filament compared to the region along the edges of mica-supported semi-cylindrical polyethylene filaments, we observed distinctly different wetting conditions, leading to the strong contrast in the thickness of the layer of molten polymers. The layer at the top of the filament, if it exists at all, is expected to be substantially thinner than in the meniscus region formed at the edges where the filament meets the mica substrate.

Consequently, the AFM phase signal measured on top of the nanofilament is dominated by the mechanical response of the underlying crystalline material, possibly with a minor (insignificant) contribution from an ultrathin surface liquid layer. In contrast, the meniscus region exhibits a much greater effective thickness. There, the AFM phase response is clearly dominated by the properties of the molten polyethylene—at least until the tip fully penetrates this region. This pronounced difference in liquid layer thickness is reflected in the strong phase contrast between the soft meniscus and the rigid crystalline filament, enabling robust and reproducible measurement of the meniscus width, which constitutes the focus of our analysis.

We appreciate the reviewer's comment and incorporated this clarification in the revised manuscript to better articulate the geometric and detection-related reasons behind the observed contrast.

The sentence "Due to the semi-cylindrical geometry of the nanofilament and its interaction with the wettable mica substrate, we have to deal with different values of the interfacial energies and surface tensions across the filament. One possibility is that not all parts of the filament are covered with a liquid layer and the conditions for surface melting (or non-melting) are not fulfilled for all parts of the nanofilament. The other possibility is that an ultrathin liquid layer exists at the top of the filament. In the latter case, it is so thin that its AFM phase signal is dominated by the mechanical response of the underlying crystalline material, possibly with a minor (insignificant) contribution from this ultrathin surface liquid layer. In contrast, the meniscus region exhibits a

much greater effective thickness. There, the AFM phase response is clearly dominated by the properties of the molten polyethylene—at least until the tip fully penetrates this region. Thus, the observed soft regions in the AFM phase images correspond to the molten polymer chains in the concave meniscus region at the edges." has been added to the paragraph after Figure 3, after the second sentence.

One could envision that entanglements that have been excluded from the growing crystal and enriched outside the crystal prevent crystallization of the liquid layer. This is invoked by some experts to explain lamellar crystallites alternating with amorphous layers in semicrystalline polymers. Is this entanglement mechanism excluded for the liquid layers observed in this study?

As discussed in our ref. (28) [Huang, D.; Hugel, T.; Reiter, G. **Mechanism for the Formation of Millimeter-Long Solid Filaments by Spin Coating Dilute Solutions of a Crystallizable Polymer**. *Macromolecules* 2024, 57 (8), 3696–3705. <https://doi.org/10.1021/acs.macromol.3c02524>.], the polymer chains within the nanofilament are highly stretched and preferentially oriented parallel to the filament axis. In the course of filament preparation via spin coating, even self-entanglements are removed efficiently. To induce filament formation, we require that the highly stretched chains, oriented and aligned in the shear flow direction, attract each other. Thus, we do not expect that there is any significant amount of entanglements between polymer chains within the filaments. Consequently, the crystal-melt coexistence phenomenon described in our manuscript occurs specifically at the lateral crystalline "growth front" of the polyethylene filaments. Due to these distinct features of the polymer chains, which mostly underwent a coil-to-stretch transition, the crystal-melt coexistence behavior is not affected by any impact of entanglements. Thus, our crystalline polymer structures are different to, for example, stacks of chain-folded lamellar polymer crystals.

On p. 2, at the very beginning of the Introduction, the authors do not present a motivation for their work. I do find it intrinsically interesting, but it would be better if a motivation or grander framework was included here.

We sincerely thank the reviewer for this constructive suggestion. We fully agree that providing a wide-ranging motivation at the beginning of the Introduction will significantly improve the

manuscript by framing our work within a broader scientific context. Thus, we have revised the opening paragraph to articulate the fundamental questions and the broader implications of studying crystal-melt coexistence, the critical size required for (secondary) nucleation and its relation to surface melting and the melting point reduction in nanoscopic polymer filaments. We hope that such a more detailed and comprehensive description of the main objectives ensures that the significance of our study becomes clear to all readers from the outset.

On page 3, the first sentence of the second paragraph has been rephrased as "In the study presented here, we experimentally explored the coexistence of a nanoscopic polymer crystal with a liquid (molten) layer characterized by a thickness, which we compared with predictions of the Lauritzen–Hoffman theory of secondary nucleation at the crystal growth front, represented by the complementary length scale of the critical nucleus. Our findings suggest that, for nanoscopic systems, the temperature dependence of the characteristic length scales of various phenomena, i.e., the size of the critical nucleus, the thickness of the liquid layer in surface premelting, and the size-dependence of the melting-point depression of finite-size crystals are related."

Figure 5c displays intriguing images that raise interesting questions. The image at top shows a very thin crystal (the bright region about 7 nm across) at high temperature – is such a small thickness compatible with the Gibbs-Thompson equation, which tells us that thin crystals melt away, in particular at high T? Have the authors raised temperature and seen the crystal melt suddenly when its thickness drops below the Gibbs-Thompson stability limit?

At the bottom of the figure, we see a completely melted filament. Observing it at gradually decreasing temperature when at some point a critical nucleus must form would seem like a worthwhile investigation, but the authors do not seem to consider critical nuclei in this relevant context.

We sincerely thank the reviewer for raising these insightful questions regarding the stability of thin crystals and the potential for real-time observation of melting and nucleation events.

As mentioned by the reviewer, the Gibbs-Thomson equation tells us that thin crystals melt away, in particular at high temperatures. In Figure 5c, we have presented this prediction of the Gibbs-Thomson equation (using a suitable prefactor) by the black line which is exactly telling that at high

temperatures the filament width w needs to be thicker than roughly two times W_{soft} . In the manuscript we wrote: “*Filaments are completely molten for $w \lesssim 2 \cdot W_{\text{soft}}(T)$.*”

Furthermore, also in Figure 5c, for a fixed temperature of $T \approx 100$ °C, we have shown that a threshold value of the size of the crystal or of the filament size w exists, as predicted by the Gibbs-Thomson equation. In the insets of Figure 5c, we showed AFM phase images for two filaments with $w \approx 26$ nm and $w \approx 42$ nm, respectively. In full agreement with the Gibbs-Thomson equation, the smaller filament was completely molten while the somewhat wider filament exhibit the remains of a tiny solid region (the reviewer measured the width of this bright region to be about 7 nm across).

We agree that ramping the temperature during a running AFM scan would be highly insightful for fundamental aspects of polymer crystallization. However, capturing melting of ultra-thin crystals upon heating or the formation of critical nuclei upon cooling is unfortunately not feasible with our current experimental setup due to inherent technical limitations. A primary constraint is the significant thermal expansion of both the sample and the heating stage during temperature changes. This expansion causes the sample surface to move vertically toward the AFM tip on the cantilever. Since our high-resolution imaging requires operating at the minimal Z-range limit of the piezoelectric scanner to maintain optimal sensitivity, even a minor amount of vertical displacement easily exceeds the available range, preventing stable imaging under these conditions.

Taking all these technical limitations into consideration, we can still confirm that the thinner filament with $w \approx 26$ nm was already molten when we reached the temperature of $T \approx 100$ °C and started the AFM measurement. Thus, within this admittedly large experimental time window, the melting process occurred.

Furthermore, as suggested by the reviewer, observing nucleation at gradually decreasing temperature is definitely a worthwhile investigation, also because theoretical works like the seminal paper by Lipowsky [R. Lipowsky, **Critical Surface Phenomena at First-Order Bulk Transitions**, Phys. Rev. Lett., 49, 1575-1578 (1982), <https://doi.org/10.1103/PhysRevLett.49.1575>] predict that the surface order parameter may behave continuously with temperature approaching the transition temperature, although the bulk

order parameter jumps at the transition temperature. Our experimental observations (considering all its inherent technical limitations, in particular with respect to its temporal resolution) suggest a continuous change in W_{soft} as temperature approaching the transition temperature. Of course, the key question raised by the reviewer on **how** nucleation occurs within a molten filament is still open. We only can state that (again accounting for the long times required for eliminating drifts induced by temperature changes) the filament with $w \approx 26$ nm, which was completely molten at the temperature of $T \approx 100$ °C, showed a crystalline core and a value of $W_{\text{soft}} \approx 9$ nm when cooled back to room temperature, demonstrating that nucleation occurred when cooling the sample.

I'm quite uncomfortable with the way in which the authors use the term "polymers" (which they do a lot, 50 times in total). To me, examples of polymers are PE, iPP, and PET. The authors use "polymers" where I would prefer "polymer chains", "polymer segments", or "polymer". Examples:

P. 3, center: "the limited number of polymers available" should probably be changed to "the limited number of polymer chains available".

P. 3, 5th line from bottom: "interactions between stretched polymers," should probably be changed to "interactions between stretched polymer chains,".

P. 4, first paragraph: "filaments of crystalline chains were surrounded initially by non-stretched molten polymers." should probably be changed to "filaments of crystalline chains were surrounded initially by non-stretched molten polymer segments."

P. 4, 2nd paragraph: "the number of available polymers is above a threshold value." Should be changed to "the number of available polymer chains [or segments?] is above a threshold value."

P. 4, below center: "sufficient mobility of noncrystallized polymers but also" should be changed to "sufficient mobility of noncrystallized polymer but also" or "sufficient mobility of noncrystallized polymer segments but also"

P. 5, second sentence: "contained molten polymers only"

Etc. etc., including

P. 18, line 7/8: “contained molten polymers which bounded the solid core region of the filament consisting of crystallized polymers.” should be changed to “contained molten polymer which bounded the solid core region of the filament consisting of crystallized polymer.”

We thank the reviewer for this precise and helpful comment regarding the use of the term "polymers." We agree that using more specific terminology such as "polymer chains" will enhance the clarity and precision of our descriptions. We have carefully revised the manuscript accordingly and replaced the general term "polymers" with the more accurate "polymer chains" throughout the text, ensuring that no unintended ambiguity is introduced.

For corrections, two different changes are made. (1), "polymer" or "polymers" is replaced by "polymer chain" or "polymer chains" when it refers to an or many polymer chains; (2), "polymers" is replaced by "polymer" when it refers to a single kind of polymer. Positions where corrections were applied are shown below:

In the following cases, the words "polymer" or "polymers" have been replaced by "polymer chain" or "polymer chains", respectively:

- P(age)3: To generate separate polymer crystals of small size, we required a system with a highly limited number of **polymer chains**.
- P3: Previous experiments²³ have shown that **polymer chains** in ultra-thin layers,
- P3: due to the limited number of **polymer chains** available in such ultra-thin films,
- P3: the homogeneous nucleation step may be circumvented by introducing filaments consisting of stretched and uniquely oriented **polymer chains**
- P3: a shear-induced change in **polymer chain** conformation can occur
- P3: i.e., **polymer chains** undergo a transition from randomly coiled to stretched chains.
- P3: attractive interactions between stretched **polymer chains**,
- P3: Due to the massive loss of conformational entropy and the high degree of order of the **polymer chains** within the filaments
- P4: **polymer chains** may crystallize easily within the filament

- P4: Thus, not all **polymer chains** can undergo such a coil-stretch transition
- P4: filaments of stretched and crystalline chains were surrounded initially by non-stretched molten **polymer chains**.
- P4: which requires that the number of available **polymer chains** is above a threshold value.
- P4: we anticipate that crystal growth will stop once the number of remaining molten **polymer chains** is below this threshold value.
- P4: allowing for sufficient mobility of non-crystallized **polymer chains** but also for crystal growth
- P4: Thus, all **polymer chains** had a chance to crystallize
- P5: If these filaments would have contained molten **polymer chains** only
- P9: Thus, as a consequence of the acting capillary forces, **polymer chains** accumulated in the wedge
- P9: The number of **polymer chains** which potentially could diffuse to the wedge-shaped meniscus region
- P9: depended on the number density of **polymer chains** adsorbed on the substrate surrounding the filament
- P9: suggests the existence of a threshold number of **polymer chains** which can be included in the meniscus region.
- P9: filaments containing stretched **polymer chains** were co-deposited together with non-stretched **polymer chains**.
- P10: formation of “kebabs” is only possible if a sufficient number of **polymer chains** is available at its growth site.
- P10: We note that an analogy but also some differences exist to observations of substrate-induced prefreezing of **polymer chains**
- P10: we observe stable coexistence of crystalline and molten **polymer chains**
- P11: we examine the origin of the coexistence of molten and crystalline **polymer chains** in such nanoscopic filaments
- P11: bead-like structures (“kebabs”) were formed only when sufficient molten **polymer chains** were available.
- P12: In order to form a stable (critical) secondary nucleus, a minimum volume of molten **polymer chains** needs to be available at the growth front of the crystal

- P12: in finite systems like ultra-thin nanometer-sized filaments the number of available molten **polymer chains** is easily exhausted
- P12: Thus, once the number of molten **polymer chains** in the surrounding
- P12: if the residual number of molten **polymer chains** at the growth front
- P12: these **polymer chains** cannot crystallize and will stay in the molten state.
- P17: even after the complete melting of the nanoscopic filaments, the stretched **polymer chains** inside the filament kept some memory which decayed only slowly

In the following cases, the word "polymers" has been replaced by "polymer":

- P4: the crystalline core was covered by a layer of molten **polymer** which was separated from the core by a well-defined interface
- P4: we analyzed the mechanical properties of filaments and their surrounding either crystalline or molten **polymer** via the phase signal of intermittent-contact mode atomic force microscopy (AFM)
- P8: within each PE filament a solid crystalline core coexisted with molten **polymer**, forming soft zones at the boundary of the filament.
- P9: the pressure p_{wedge} inside this wedge of molten **polymer** was lower than the surrounding atmospheric pressure p_{atm} ,
- P9: The latter formed an adsorbed ultrathin layer of liquid-like **polymer** in the surrounding of the filaments.
- P12: we anticipate that filaments with a crystalline core (blue area in Figure 3a) are surrounded with a narrow zone of molten **polymer** at the growth front
- P12: we arrive at two predictions for the equilibrium width W_{soft} of the soft zone of molten **polymer**
- P12: we assume that a cylindrical secondary nucleus can only form in the soft zone of molten **polymer** at the periphery of a hard core
- P13: we may assume that the geometry of the soft meniscus region containing molten **polymer** is the same for all filaments

- P15: we can conclude that all filaments, including filaments decorated with bead-like structures, were bounded by coexisting soft zones of molten **polymer** of a size just below a size proportional to $r^*(\Delta T)$ of the critical secondary nucleus.
- P16: Besides the limiting size derived from the minimum volume of molten **polymer** required for the formation of a stable (critical) secondary nucleus at the crystal growth front
- P17: distinguishing even small differences in elastic properties of **polymer**.
- P17: we conclude that the soft region contained molten **polymer** which bounded the solid core region
- P17: the solid core region of the filament consisting of the crystallized **polymer**.
- P18: If the size of the soft zone of molten **polymer**, characterized by its width $W_{\text{soft}}(T)$,

The text contains various other flaws; here are some examples:

We are deeply grateful to the reviewer for the meticulous reading of our manuscript and for providing such a comprehensive list of specific corrections. We agree with all the points raised and have carefully revised the text accordingly to address each of the flaws highlighted, including the rephrasing of awkward constructions, clarification of ambiguous statements, correction of grammatical errors and typos, and consistent use of precise terminology.

- *P. 1, first line of the Abstract: “is causing that crystals are often” Please rephrase. E.g. “results in crystals often being”*

We have changed that.

- *P. 2 below center: “several theoretical approaches” could the authors name some of them here?*

There were references behind the dot of this sentence as examples.

(11) Lipowsky, R.; Speth, W. Semi-Infinite Systems with First-Order Bulk Transitions. *Phys. Rev. B* **1983**, 28 (7), 3983–3993. <https://doi.org/10.1103/PhysRevB.28.3983>.

(12) Conde, M. M.; Vega, C.; Patrykiewicz, A. The Thickness of a Liquid Layer on the Free Surface of Ice as Obtained from Computer Simulation. *The Journal of Chemical Physics* **2008**, 129 (1), 014702. <https://doi.org/10.1063/1.2940195>.

- *P. 2 last paragraph, “for a finite number N of molecules and a finite volume V”: I believe the authors mean “for a fixed number N of molecules and a fixed volume V”*

We have changed that.

- *P. 3 near the top, “and T are constant”: It would be better to add “while the particle number N can vary/fluctuate” for clarity.*

The sentence has been changed to “where the chemical potential μ , V , and T are constant **while the number N of particles in the microstates can vary**, is unstable and has been identified as the critical nucleus.”

- *P. 3 second paragraph: “In the here presented study” needs to be changed to “In the study presented here”. Similarly, for “here applied conditions” on p. 7, “The here investigated filaments” on p. 8, and “here investigated filaments” on p. 10.*

We have changed all these points.

- *P. 3 in the 2nd paragraph, “polymers in ultrathin layers ... crystallized very slowly.” I think this should be changed to present tense: “polymers in ultrathin layers ... crystallize very slowly.”*

We have changed that.

- *P. 5, third sentence starting with “Naively”. It should be made clear that the “uniform mechanical properties” should be considered across rather than along the filaments.*

We have included this useful information in the following sentence, which helps clarify that the difference in mechanical properties is perpendicular to the filament.

The sentence “*However, the phase signal derived from the intermittent-contact mode AFM measurements clearly revealed the coexistence of two regions **within** the filaments, characterized by distinctly different properties.*” has been changed to “*However, the phase signal derived from the intermittent-contact mode AFM measurements clearly revealed the coexistence of two regions **across** the filaments, characterized by distinctly different **mechanical** properties.*”

- *P. 5, captions of Figure 5: “or a PE-B”.*

The sentence “**The PE-M sample was used here.**” has been added to the figure caption of Figure 5.

- *P. 5, 2nd line from the bottom can be made clearer: “For an easier comparison of a) with b) and c) with d),”*

We have changed that.

- *P. 6: “in the Methods” section.”*

We have changed that.

- *P. 8 right above Figure 3: “coexisted with molten EDGES, soft zones at the LATERAL boundaries of the filament.”*

The sentence “PE filament a solid crystalline core coexisted with molten polymer, forming soft zones at the boundary of the filament.” has been changed to “PE filament a solid crystalline core coexisted with **meniscus edges** of molten polymer, **i.e.**, soft zones at the **lateral** boundary of the filament.”

- *P. 9: The tense of verbs in the first paragraph should be changed to present tense.*

The verb tense in that paragraph has been revised to the present tense for sentences that describe general facts or established knowledge.

- *P. 9, beginning of second paragraph: “SOME with a uniform width and OTHERS decorated”.*

We have changed that.

- *P. 13 top: What is the basis of “the assumption of $W_{\text{soft}} \sim r^*$ ”?*

“Moreover, since secondary nuclei adopt the same orientation as the crystal face on which they are formed — that is, polymer chains are oriented and aligned along the axis of the crystalline nanofilament — a secondary cylindrical nucleus is also aligned along the axis of the nanofilament. There, the radius is in the direction perpendicular to the axis of the filament. Accordingly, we assume a relation between the critical radius for nucleation (r^*) with the value of W_{soft} , *i.e.*, $W_{\text{soft}} \sim r^*$.” has been added to P12 of the manuscript before the sentence “Consequently, we anticipate that filaments with a crystalline core (blue area in Figure 3a) are surrounded with a narrow zone of molten polymer at the growth front (red area in Figure 3a).”

- *P. 13, 4th line: “polymer is not expected to influence neither r^* nor W_{soft} .” contains too many negatives. Correct: “polymer is expected to influence neither r^* nor W_{soft} .” or “polymer is not expected to influence either r^* or W_{soft} .”*

We have changed that.

- *P. 13, below eq.(1): “ ΔG_V is the free energy difference involved in the formation of a crystal nucleus”. Dimensional analysis indicates that “ ΔG_V is the free energy change, per volume,” of the formation of a crystal nucleus”. We have changed that.*

- *P. 13 below eq.(2): “ ρ_n is the number density of molecules” is incompatible with later “ ρ_n is ranging from ca. 780 kg/m³ to ca. 970 kg/m³”, which shows that “ ρ_n is the mass density of the nucleus”. P. 13 below eq.(2): “ ΔH_f is the heat of fusion per volume element” is incompatible with “ $\Delta H_f = 4.1 \pm 0.2$ kJ/mol of CH₂”. Dimensional analysis indicates that): “ ΔH_f is the heat of fusion per mass” or “ ΔH_f is the specific heat of fusion”.*

The sentence “ ρ_n is the number density of molecules, ΔH_f is the heat of fusion per volume element of polymer at $T_m^\infty = 141$ °C.” has been changed to “ ρ_n is the mass density of the polymer, ΔH_f is the heat of fusion per unit mass of the polymer at $T_m^\infty = 141$ °C. ”

- *P. 13: I would propose to write “nm” at the end of the equation instead of the awkward “in [nm]”.*

Equation (3) has been changed to:

$$r^*(\Delta T) \approx P \cdot \frac{(T_m^\infty)^2}{T \cdot \Delta T} \quad (3)$$

where the prefactor $P = \frac{2\sigma_s}{\rho_m \cdot \Delta H_f} = (0.1 \pm 0.02)$ nm.

For calculating P , we used the σ_s is ranging from 12 mJ/m² to 15 mJ/m²; $\Delta H_f \approx 4.1 \pm 0.2$ kJ/mol of CH₂, and ρ_m is ranging from ca. 780 kg/m³ to ca. 970 kg/m³.

- *P. 14, “on various filaments, either they had a”: Better: “on various filaments, with either a”*
We have changed that.
- *P. 15, Figure 5: In part c), it would be good to mark the absolute T values on the top axis, to counter the impression that W decreases with increasing temperature.*
We have changed that.
- *P. 15, Figure 5: The font sizes are too disparate. They should be made more uniform (mostly by increasing the size of the small lettering).*
We have changed that.
- *P. 15, caption of Figure 5: The sentence starting with “The two AFM phase images” seems to lack a main verb.*
The two AFM phase images were measured at $T \approx 100$ °C, ...
- *P. 16, right below the table: “and possible intercepts.” should probably be rephrased as “and additive constants.”* We have changed that.
- *P. 17: “Spin coating experiments were performed”* We have changed that.

Detailed response on comments (*written in blue*) from Reviewer #3

This paper presents an interesting study on the coexistence of a crystal- melt interface on supported polyethylene filaments. I found that the paper is not easy to read, and at times it is confusing. I urge the authors to revise it and make it more understandable.

We sincerely thank Reviewer #3 for the invested time and the careful evaluation of our manuscript. We appreciate the positive assessment of our study judging it as interesting and acknowledge the constructive feedback regarding clarity. We agree that the manuscript, in its previous form, was not easy to read and might have been confusing at times. We apologize for these shortcomings and undertook a comprehensive revision to significantly improve the organization, flow, and overall readability of the text. We now present our findings in a clearer, more logical, and accessible manner to ensure the message is effectively communicated to the reader.

The first phrase of the abstract is quite cryptic and not well written: "The broken symmetry of molecular interactions at interfaces is causing that crystals are often covered by a thin liquid layer of their own melt." No explanation is given to what they refer as broken symmetry and they cannot use the verb in this tense "is causing", as this denotes that is happening at this instant...Consider revising and rewriting the abstract to indicate clearly what has been done and what is new and a significant advanced with respect to existing literature.

We thank the reviewer for this critical comment and for highlighting the lack of clarity in the opening sentence of our abstract. We agree that the original phrasing was cryptic and that the verb tense was inappropriate. We aimed to refer to the symmetry of the interactions with neighboring molecules which is different for molecules in the bulk and molecules at interfaces. When going from a crystalline phase to the surrounding phase (liquid or gas), several symmetry elements change, for example, from a three-dimensionally ordered to a disordered state of translational invariance. One may consider an interface or surface as a planar perturbation reflected by changes in (number) density, (two-or three-dimensional) order or translational/rotational symmetry in the perpendicular direction. For systems governed by short-range forces, such a perturbation decays exponentially. Long range interactions typically cause a slower decay of these perturbations. Inspired by the idea of broken symmetry introduced by Anderson (Anderson PW. 1972 **More is**

different. Science 177, 393–396; doi:10.1126/science.177.4047.393) where he described the scale-dependent limitations of symmetrical physical laws for explaining common macromolecular structures, various theoretical treatments have been developed to account for the properties of crystal surfaces leading to concepts of surface melting, premelting or quasi-liquid (molten) layers (see for example: R. Lipowsky, **Critical Surface Phenomena at First-Order Bulk Transitions**, Phys. Rev. Lett., 49, 1575 (1982), <https://doi.org/10.1103/PhysRevLett.49.1575> , P. Nozières, **Surface melting and crystal shape.** J. de Phys., 1989, 50 (18), pp.2541-2550.<https://doi.org/10.1051/jphys:0198900500180254100> ; U. Tartaglino, T. Zykova-Timan, F. Ercolessi, E. Tosatti, **Melting and nonmelting of solid surfaces and nanosystems**, Phys. Rep. 411 (2005) 291–321, <https://doi.org/10.1016/j.physrep.2005.01.004>) and appropriate experiments were performed (see for example: Joost W. M. Frenken, Peter M. J. Maree, and J. Friso van der Veen, **Observation of surface-initiated melting**, Physical Review B, 34(11), 7506-7516 (1986); <https://doi.org/10.1103/PhysRevB.34.7506> ; J. G. Dash (1989) **Surface melting**, Contemporary Physics, 30:2, 89-100, <https://doi.org/10.1080/00107518908225509> ; Yang Yang, Mark Asta, and Brian B. Laird, **Solid-Liquid Interfacial Premelting**, Phys. Rev. Lett. 110, 096102 – Published 28 February, 2013; <https://doi.org/10.1103/PhysRevLett.110.096102> ; Gen Sazaki, Harutoshi Asakawa, Ken Nagashima, Shunichi Nakatsubo, and Yoshinori Furukawa, **How do Quasi-Liquid Layers Emerge from Ice Crystal Surfaces?** Cryst. Growth Des. 13, 1761–1766 (2013), <https://doi.org/10.1021/cg400086j>).

Thus, in order to avoid confusion by trying to capture the complexity of the problem in a short sentence, we have revised the opening sentence of our abstract. It reads now as: “An interface or surface may be considered as a planar perturbation reflected by changes in molecular properties in the direction perpendicular to the interface ore surface. As a consequence, predicted by theory and shown by experiments, crystals are often covered by a thin liquid layer of their own melt.” Furthermore, we undertook a comprehensive revision of the entire abstract to ensure it clearly states the objectives, findings, and significant advances of our work with respect to the existing literature.

They write: For all studied PE filaments and independent of their size, the coexistence of a hard (solid) core bounded by a soft (liquid-like) layer near the filament-substrate contact line was confirmed.

Are they sure this holds for any PE filament size? Are we to expect a molten surface in a high molecular weight microscopic-sized PE fiber or bundles of fibers to have the same molten surface. It would seem that the phenomena observed in most likely present in nanometer fibers.

We thank the reviewer for this important question, which allows us to clarify the scope of our claim. In agreement with theoretical predictions and supported by experiments on various systems, including semi-infinite metal surfaces and ice crystals, we believe that our statement that “a soft, liquid-like meniscus layer coexists with a solid core” essentially applies in general and is not limited by size or geometry. In our specific investigation, we have verified that all studied PE filaments ranging from 10 nm to 100 nm were bounded by an approximately constant soft (liquid-like) layer near the filament-substrate contact line, with a comparable width, W_{soft} , for all filaments. Furthermore, the same soft layer was observed at the periphery of the much larger, bead-like structures that decorated some of these nanofilaments and reached a size up to the micrometer scale. However, while we believe that all polymer crystals are bounded by a nanoscopic liquid layer, we anticipate that several factors like geometry and chain folds do have an impact on (and will lead to changes of) the observed value of W_{soft} . Therefore, we agree that our conclusions based on the observations of the specific system of PE nanofilaments cannot be generalized without precautions and deviations in the value of W_{soft} have to be expected. In particular, for different geometries like lamellar crystals of highly folded crystalline states of polymers we need to perform further dedicated investigations.

The sentence "We notice that, for different geometries like lamellar crystals of highly folded crystalline states of polymers, further dedicated investigations need to be performed." has been added at the end of the first paragraph after Figure 1.

In many instances, the authors use the word "polymer" to refer to polymer chains, for instance, just one example:

"However, in finite systems like ultra-thin nanometer-sized filaments the number of available molten polymers is easily exhausted. Thus, once the number of molten polymers in the surrounding and at the growth front is below this minimum value, we anticipate that crystal growth will stop." There is no way they can use the expression "number of available molten polymers is easily exhausted." I imagine they are referring to the available molten chains; otherwise, I do not understand their meaning. This happens several times in the paper.

We thank the reviewer for this precise and helpful comment regarding the use of the term "polymers." We agree that using a more specific terminology such as "polymer chains" enhances the clarity and precision of our descriptions. We have carefully revised the manuscript accordingly and replaced, whenever appropriate, the general term "polymers" with the more accurate "polymer chains" throughout the text. We hope that we have removed all unintended ambiguity.

- P(age)3: To generate separate polymer crystals of small size, we required a system with a highly limited number of **polymer chains**.
- P3: Previous experiments²³ have shown that **polymer chains** in ultra-thin layers,
- P3: due to the limited number of **polymer chains** available in such ultra-thin films,
- P3: the homogeneous nucleation step may be circumvented by introducing filaments consisting of stretched and uniquely oriented **polymer chains**
- P3: a shear-induced change in **polymer chain** conformation can occur
- P3: i.e., **polymer chains** undergo a transition from randomly coiled to stretched chains.
- P3: attractive interactions between stretched **polymer chains**,
- P3: Due to the massive loss of conformational entropy and the high degree of order of the **polymer chains** within the filaments
- P4: **polymer chains** may crystallize easily within the filament
- P4: Thus, not all **polymer chains** can undergo such a coil-stretch transition
- P4: filaments of stretched and crystalline chains were surrounded initially by non-stretched molten **polymer chains**.
- P4: which requires that the number of available **polymer chains** is above a threshold value.
- P4: we anticipate that crystal growth will stop once the number of remaining molten **polymer chains** is below this threshold value.
- P4: allowing for sufficient mobility of non-crystallized **polymer chains** but also for crystal growth
- P4: Thus, all **polymer chains** had a chance to crystallize
- P5: If these filaments would have contained molten **polymer chains** only
- P9: Thus, as a consequence of the acting capillary forces, **polymer chains** accumulated in the wedge
- P9: The number of **polymer chains** which potentially could diffuse to the wedge-shaped meniscus region

- P9: depended on the number density of **polymer chains** adsorbed on the substrate surrounding the filament
- P9: suggests the existence of a threshold number of **polymer chains** which can be included in the meniscus region.
- P9: filaments containing stretched **polymer chains** were co-deposited together with non-stretched **polymer chains**.
- P10: formation of “kebabs” is only possible if a sufficient number of **polymer chains** is available at its growth site.
- P10: We note that an analogy but also some differences exist to observations of substrate-induced prefreezing of **polymer chains**
- P10: we observe stable coexistence of crystalline and molten **polymer chains**
- P11: we examine the origin of the coexistence of molten and crystalline **polymer chains** in such nanoscopic filaments
- P11: bead-like structures (“kebabs”) were formed only when sufficient molten **polymer chains** were available.
- P12: In order to form a stable (critical) secondary nucleus, a minimum volume of molten **polymer chains** needs to be available at the growth front of the crystal
- P12: in finite systems like ultra-thin nanometer-sized filaments the number of available molten **polymer chains** is easily exhausted
- P12: Thus, once the number of molten **polymer chains** in the surrounding
- P12: if the residual number of molten **polymer chains** at the growth front
- P12: these **polymer chains** cannot crystallize and will stay in the molten state.
- P17: even after the complete melting of the nanoscopic filaments, the stretched **polymer chains** inside the filament kept some memory which decayed only slowly

In the following cases, the word "polymers" has been replaced by "polymer":

- P4: the crystalline core was covered by a layer of molten **polymer** which was separated from the core by a well-defined interface

- P4: we analyzed the mechanical properties of filaments and their surrounding either crystalline or molten **polymer** via the phase signal of intermittent-contact mode atomic force microscopy (AFM)
- P8: within each PE filament a solid crystalline core coexisted with molten **polymer**, forming soft zones at the boundary of the filament.
- P9: the pressure p_{wedge} inside this wedge of molten **polymer** was lower than the surrounding atmospheric pressure p_{atm} ,
- P9: The latter formed an adsorbed ultrathin layer of liquid-like **polymer** in the surrounding of the filaments.
- P12: we anticipate that filaments with a crystalline core (blue area in Figure 3a) are surrounded with a narrow zone of molten **polymer** at the growth front
- P12: we arrive at two predictions for the equilibrium width W_{soft} of the soft zone of molten **polymer**
- P12: we assume that a cylindrical secondary nucleus can only form in the soft zone of molten **polymer** at the periphery of a hard core
- P13: we may assume that the geometry of the soft meniscus region containing molten **polymer** is the same for all filaments
- P15: we can conclude that all filaments, including filaments decorated with bead-like structures, were bounded by coexisting soft zones of molten **polymer** of a size just below a size proportional to $r^*(\Delta T)$ of the critical secondary nucleus.
- P16: Besides the limiting size derived from the minimum volume of molten **polymer** required for the formation of a stable (critical) secondary nucleus at the crystal growth front
- P17: distinguishing even small differences in elastic properties of **polymer**.
- P17: we conclude that the soft region contained molten **polymer** which bounded the solid core region
- P17: the solid core region of the filament consisting of the crystallized **polymer**.
- P18: If the size of the soft zone of molten **polymer**, characterized by its width $W_{\text{soft}}(T)$,

The experiments performed and the analyses presented are very good, but unfortunately, the paper is in many parts not clear enough, and the writing should be improved.

We sincerely thank the reviewer for the positive assessment of our experiments and analyses, and for the constructive feedback regarding the clarity of our manuscript. We acknowledge that, in order to enhance readability, the writing of many parts of the text required significant improvement. We have undertaken a thorough revision of the entire manuscript, focusing on improving the organization, flow, and clarity of the text, ensuring that our findings are presented in a more logical, coherent, and accessible manner.

They have employed only PE filaments, but in principle, they could have used other filaments where they can easily vary the degree of crystallinity and check its influence (i.e., PET). The intermolecular interactions in this case are weak Van Der Waal forces (as only PE was employed). What would have happened if a polymer like PEO is used or PCL, in which the intermolecular forces are stronger?

We thank the reviewer for this insightful comment and for suggesting the possibility of extending our study to other polymer systems with varying degrees of crystallinity and stronger intermolecular interactions, such as PET, PEO, or PCL. We agree that investigating these materials will provide valuable comparative insights into the role of polymer chemistry and intermolecular forces on surface premelting and crystal-melt coexistence.

However, the goal of our present manuscript was (i) to verify the coexistence of a crystalline polymer filament with a bounding layer of molten polymers, (ii) to demonstrate that the thickness (width W_{soft}) of this layer is independent of size, molecular weight or (some) geometrical features of the filament like the “beads-on-a-string” morphology, and (iii) to show that W_{soft} is increasing as the temperature is raised. The comparison of the temperature-dependence of W_{soft} with predictions of various theoretical concepts suggests or even indicates that these concepts may be closely related, as we discuss in our manuscript.

Of course, in subsequent works, we aim to systematically apply the same methodology to a broader range of polymers to precisely explore the influences of crystallinity and intermolecular force strength/range on the phenomena described here, but also to explore the impact of different geometries

like lamellar crystals of highly folded crystalline states of polymers on values of W_{soft} . Thus, the reviewer's suggestion provides excellent guidance for, and is fully in line with, the direction of our future research.

Is there any molecular weight dependence of the effects observed?

We thank the reviewer for raising this important question regarding the potential molecular weight dependence of the observed effects. In our manuscript we addressed this point indirectly, by comparing samples of polyethylene (samples **PE-M** and **PE-B**) consisting of chains of different lengths and of extremely different dispersity. For these samples, we systematically examined the width of the soft layer (W_{soft}) across polyethylene nanofilaments, as summarized in Figure 4. Our analysis revealed that, across all the samples of different molecular weights studied, the values of W_{soft} remained consistently within the bounds of experimental uncertainty. This indicates that, under the conditions of our experiments, the phenomenon of crystal-melt coexistence, as characterized by the value of W_{soft} , is independent of molecular weight.

Comment #1 of Reviewer #2: Despite their long responses to the reviews, the authors have actually only made few changes to the manuscript, and I consider it still significantly deficient.

Response of the authors: We are grateful to the reviewer for giving us the opportunity to re-emphasize the **RELATION** between the three phenomena, nucleation, surface premelting and the melting point depression of finite size crystals. We now explicitly write how the broken symmetry of intermolecular interactions at interfaces introduces a “common” length scale through the competition of interfacial energy (“surface term”) and internal energy (“volume term”). However, we respectfully disagree with the assessment that our “*manuscript is still significantly deficient*” and that we “*only made few changes to the manuscript*”.

As documented in the “**Highlighted Changes**” of our manuscript and the Supporting Information as well in our point-by-point response to the reviewers, we have made extensive revisions. These include: changes in the Abstract; a strengthened Introduction; clarification of theoretical frameworks; addition of new experimental data and analysis in the Supplementary Information (e.g., Section 8, Fig. SI-5 with added discussion in Section 2); systematic clarification of terminology throughout the text (over 40 instances); and direct responses to all technical and theoretical concerns raised by all reviewers. We, together with reviewer #1 and reviewer #3, trust that our response represented substantial improvements and believe that we adequately answered all questions.

Comment #2 of Reviewer #2: I thought that the experimental results were interesting, but Reviewer 1 has cast doubts on those.

Response of the authors: We thank the reviewer for finding our results interesting. We note that Reviewer #1, in her/his final comment on the revised manuscript, stated: “*In their revised manuscript and its supplementary information, the authors responded nicely to my concerns by giving more throughout discussions and additional figures. Thus, I am now positive for the revised version to be accepted in this journal.*” This indicates that the initial questions of Reviewer #1 were fully resolved through our revisions and additions. Thus, the “doubts” of Reviewer #2 are not substantiated by Reviewer #1.

Comment #3 of Reviewer #2: I consider the theory to be sloppy and unconvincing.

Response of the authors: We note that the reviewer's criticism appears to be focused on established theoretical concepts (nucleation, surface premelting, and the melting point depression of finite size crystals), rather than our application of these concepts, which are cornerstones of the science of crystallization, developed and refined over decades by the scientific community and supported by a vast body of literature.

In our manuscript, our goal was not to validate these established theories, but to use them to interpret our experimental observations. We have demonstrated in our manuscript that the classical nucleation theory provides a consistent interpretation for the observed crystal-melt coexistence in supported PE nanofilaments. This interpretation is further strengthened by our experimental confirmation of the temperature dependence predicted by classical nucleation theory, as detailed in our manuscript (Pages 12-16).

The reason for involving the other two phenomena is twofold. First, they share the same fundamental physical origin with nucleation: the competition between a surface energy term and a volume energy term. Second, the characteristic length scales derived from all three theories, *i.e.*, $\ell \sim$ the ratio of "surface term"/"volume term", are compatible with our experimental data for $W_{\text{soft}}(T)$. By showing that our experimental data can be described by the various temperature dependences of $\ell(T)$ derived for these three distinct phenomena, we pointed out a possible relationship. This link to the ratio of "surface term"/"volume term", therefore, serves as a compelling starting point for future investigations.

Therefore, we believe a general critique of these fundamental theories is not justified. Regardless of the intended focus of the criticism, we maintain that our specific application of these theories to interpret our experimental data is both appropriate and insightful. We and the other two reviewers could not identify anything “*sloppy and unconvincing*” in these concepts and their theoretical descriptions.

Comment #4 of Reviewer #2: The authors still claim, in the Abstract, that they “show that these three phenomena are related”, while they really don’t (which they even admitted in their responses: “We agree that in our manuscript we only indicated/suggested the possibility of unifying various concepts, i.e., the concepts of nucleation, surface premelting, and finite-size melting point depression. Of course, a thorough proof of such unification requires a profound theoretical development, which is beyond the scope of this study.”).

Response of the authors: There is no contradiction. As we have explicitly and repeatedly written in our manuscript, for example, at the end of the Results section (page 18): “*all three phenomena originate from the broken symmetry of intermolecular interactions at interfaces and the corresponding competition between interfacial energy and internal energy.*” It is this competition of interfacial energy (“surface term”) and internal energy (“volume term”) that **RELATES** all three phenomena. Over the last 50 years and more, the interplay of a “surface term” and a “volume term” has been identified in all experimental studies and introduced in all theoretical descriptions of nucleation, surface premelting and the melting point depression of finite size crystals. As the reviewer certainly knows very well, the balance of an energy per surface (the “surface term”) and an energy per volume (the “volume term”), expressed by the ratio of these two terms, introduces a **length scale** “ ℓ ”. The ratio of “surface term”/“volume term” is at the basis of the size of the critical nucleus (r^*), reflects the thickness (l) of the liquid layer in surface premelting and determines for crystals of a finite size (σ) their melting temperature, *i.e.*, σ is the length scale that controls the amount of melting point depression. All existing theoretical descriptions of these three phenomena yield a **temperature dependence** of this length scale “ ℓ ”. In our manuscript, we have compared the predictions of the temperature dependence of all three length scales (*i.e.*, r^* , l and σ) with our experimentally observed values of $W_{\text{soft}}(T)$. This comparison demonstrated that, within the uncertainty of the measured values of $W_{\text{soft}}(T)$, all three phenomena are compatible and consistent with our observations. In our opinion, this should not surprise the reviewer because all these phenomena are reflecting the competition of interfacial energy (“surface term”) and internal energy (“volume term”).

We stated in the Abstract: “*Filaments smaller than ca. $2 \cdot W_{\text{soft}}(T)$ were completely molten. The values of $W_{\text{soft}}(T)$ compared favorably with theoretically predicted characteristic length scales*”

in the context of nucleation, surface premelting and the melting point depression of finite size crystals.” This is an observation of an empirical relationship, not a claim of theoretical unification. Our admission that (in page 17 of our previous reply to reviewers) *“a thorough proof of such unification requires a profound theoretical development, which is beyond the scope of this study”* precisely acknowledges that. By focusing on the temperature dependence, we are presenting a relation between experimental observations and several theoretical concepts, but not a theoretical proof. A detailed theoretical description would require to account for the various parameters characterizing the geometry of the object (the shape of the filament on the substrate) and the various molecular interactions, often summarized in the interplay of interfacial tensions at the three phase contact line.

Comment #5 of Reviewer #2: In that case it is unacceptable to write “we show that these three phenomena are related”.

Response of the authors: As we have explicitly and repeatedly written in our manuscript, for example at the end of the Results section (page 18): *“all three phenomena originate from the broken symmetry of intermolecular interactions at interfaces and the corresponding competition between interfacial energy and internal energy.”* It is this competition of interfacial energy (“surface term”) and internal energy (“volume term”) that **RELATES** all three phenomena. The balance of an energy per surface (the “surface term”) and an energy per volume (the “volume term”), expressed by the ratio of these two terms, introduces a **length scale** “ ℓ ”. Therefore, we believe it is entirely acceptable to write that *“we show that these three phenomena are related”*.

Demonstrating that one experimental dataset can be described by the functional forms of three different theoretical phenomena reflects the relationship between them, specifically as they are governed by same underlying physics (the acting intermolecular forces), which is represented by the ratio of “surface term”/“volume term”. In our manuscript, we have written:

In the Abstract: *“Altogether, we show that these three phenomena are related and dominated by the intermolecular forces acting at crystal surfaces.”*

At the end of the Results section (Page 18): *“Although these phenomena were often treated independently, our results highlight their fundamental relationship. All three phenomena originate*

from the broken symmetry of intermolecular interactions at surfaces and the corresponding competition between interfacial energy and internal energy.”

At the end of the Discussion section (Page 19): *“While the physical concepts underlying these three phenomena are different, they all were capable to describe our results of $W_{soft}(T)$ within the uncertainty of the datapoints, highlighting the fact that all these phenomena originate from the same intermolecular interactions at the surface of crystals.”*

Comment #6 of Reviewer #2: Just because two phenomena have a similar simple T-dependence doesn't show that they are related.

Response of the authors: We agree that similar functional forms **alone** should not be considered as a conclusive proof of a relation between different physical (theoretical) concepts. However, as described in our answer to comment #4 of Reviewer #2, the three phenomena are governed by the same underlying physical interactions (intermolecular forces), typically represented by the ratio of “surface term”/“volume term”.

Comment #7 of Reviewer #2: The theory remains at a disappointingly low level, mostly just citing diverse sources (see Table 1 and the scant text below it).

Response of the authors: As expressed through the title of the manuscript, *i.e.*, *“Temperature dependence of crystal – melt coexistence for supported polyethylene filaments”*, our central objective is to interpret the temperature dependence of the experimentally observed coexistence of a crystalline and a liquid phase. To this end, we compared our experimental results with predictions of well-established existing theoretical concepts [ref. 6-10 of the manuscript], all incorporating the competition between interfacial energy and internal energy. The essence of these theories can be found in textbooks. [Book1, Book2] We therefore did not repeat all details of these theories in our manuscript.

[Book1] Muthukumar, M. Nucleation in Polymer Crystallization. In *Advances in Chemical Physics*; John Wiley & Sons, Ltd, 2003; pp 1–63. <https://doi.org/10.1002/0471484237.ch1>.

[Book2] Reiter, G.; Strobl, G. R. Progress in Understanding of Polymer Crystallization; Springer Science & Business Media, 2007.

Comment #8 of Reviewer #2: The title is also still inaccurate -- the authors simply refused to change it to “Temperature dependence of liquid edge layers of supported polyethylene filaments”.

Response of the authors: We respectfully declined the suggestion because we believe the original title is more accurate and conceptually significant. In the manuscript, we have explained why a crystal can coexist (in equilibrium) with a liquid layer on its surface. Relying on published works, we also mentioned how the thickness of this layer changes with temperature, and why this thickness is related to the ratio of “surface term”/“volume term”.

Besides, in our previous reply to reviewers, we have provided additional arguments. On Page 16 of our previous reply to reviewers, we have written: *“We sincerely appreciate the reviewer’s thoughtful suggestion regarding the title of our manuscript. We agree that clarity and accuracy are essential. The proposed alternative title, “Temperature Dependence of Liquid Edge Layers of Supported Polyethylene Filaments,” certainly captures our key experimental observation. At the same time, we believe that the original title, “Temperature Dependence of Crystal–Melt Coexistence for Supported Polyethylene Filaments,” is better suited because it captures in addition the essential physical phenomenon underlying the formation of the liquid edge layers—namely, the equilibrium coexistence between crystalline and molten phases characterized by a length scale which depends on temperature”.*

Comment #9 of Reviewer #2: The authors also have not changed the misleading claim that Table 1 shows “the characteristic length scale, represented as l , of a liquid layer on the surface of a crystal, as predicted by three different phenomena”, while at least two of the entries are actually the limiting thicknesses of crystals.

Response of the authors: In our manuscript and in our previous reply to the reviewers, we have addressed carefully how the thickness of a molten layer on the surface of the crystal is linked with the size of a critical nucleus and the melting point depression of a crystal of finite size. To clarify this also in the caption of the Table 1, we have now specified in this caption that details are given in the main text (see answer to Comment #10 of Reviewer #2). Providing all information in the

caption would be inappropriate. Here, we present a summary of the explanations given in the main text:

In our manuscript on page 13, we explained the relation of W_{soft} with the size of the critical nucleus, as predicted by classical nucleation theory (CNT): *“Moreover, since the secondary nuclei adopt the same orientation as the original crystal—that is, the nanofilament in which the polymer chains are aligned along its axis—the cylindrical nucleus is aligned lengthwise along the nanofilament, while its radius is oriented perpendicularly; this radius is directly correlated with the value of W_{soft} , i.e., $W_{\text{soft}} \sim r^*$.”*

In our manuscript on page 17, we related W_{soft} with the melting point depression of a crystal of finite size: *“Interestingly, for a given finite size of a filament, characterized by its width w , such an increase of $W_{\text{soft}}(T)$ allows to define a size-dependent melting temperature of a filament $T_m(w) \leq T_m(w = \infty) = T_m^\infty$, a phenomenon often referred to as size-dependent “melting point depression” of nanostructures.^{7,8} In other words, when $2 \cdot W_{\text{soft}}(T) \approx w$, then the whole filament is molten.”*

In our manuscript on page 17, we also linked W_{soft} with the coexistence of a liquid layer on a crystal: *... the coexistence of a liquid layer on a crystal is also theoretically predicted^{6,11,13} and experimentally verified^{6,11,13} for planar surfaces on bulk samples in the context of surface premelting. Assuming a specific distance-dependence of intermolecular interactions, various functional forms of the temperature variation of the characteristic length scale, represented as l , of a liquid layer on the surface of a crystal have been predicted (see Table 1). However, these approaches did not account for the influence of geometry and curvature, making a direct comparison with our results problematic.*

In addition, in our previous reply to the reviewers, we have written on Page 18: *“As described in a large number of publications and also demonstrated in recent computer simulations on nanoscopic systems, in equilibrium, a crystal can be covered by a liquid layer of its melt. As shown, for example, in ref (1) of our manuscript, U. Tartaglino, T. Zykova-Timan, F. Ercolessi, E. Tosatti, **Melting and nonmelting of solid surfaces and nanosystems**, Phys. Rep. 411 (2005) 291–321, <https://doi.org/10.1016/j.physrep.2005.01.004>, the thickness $l(T)$ of this liquid layer can be*

*related to the wetting behavior described via the corresponding surface and interfacial tensions. For curved interfaces, one has to include additional terms like, for example, a term characterized by the Tolman length (see, e.g., P. Montero de Hijes, Jorge R. Espinosa, Valentino Bianco, Eduardo Sanz, and Carlos Vega, **Interfacial Free Energy and Tolman Length of Curved Liquid-Solid Interfaces from Equilibrium Studies**, *J. Phys. Chem. C*, 124, 8795-8805 (2020); <https://doi.org/10.1021/acs.jpcc.0c00816>). Represented through an “interaction” free energy term $V(l)$, which includes the various short- and long-range intermolecular forces at work, the solid–liquid and the liquid–vapor interfaces “feel” their mutual presence. As described through equation (2) in this above-mentioned paper by Tartaglino et al., the thickness $l(T)$ of this layer diverges as the melting temperature is approached. Thus, a crystal of finite size σ will be molten completely when the molten layer covers the whole size of the crystal., i.e., $l(T) \geq \sigma$. Consequently, similar to what we showed in Fig. 5 of our manuscript, the value of $l(T)$ decides also the melting temperature of finite size crystals.”*

Comment #10 of Reviewer #2: The response claims “regarding the interpretation of characteristic length scales in Figure 5c and Table 1. We agree that the caption may have been misleading in its presentation. We have thus improved its clarity.” but there is no pertinent yellow highlighted change indicated in either caption (and the caption of Table 1 does remain misleading). This seems deceptive.

Response of the authors: We apologize that, according to Reviewer #2, *the caption of Table 1 does remain misleading.*

We have now explicitly repeated in this caption that the balance of an energy per surface (the “surface term”) and an energy per volume (the “volume term”), expressed by the ratio of these two terms, introduces a **length scale** “ ℓ ”. The ratio of “surface term”/“volume term” is at the basis of the size of the critical nucleus (r^*), reflects the thickness (l) of the liquid layer in surface premelting and determines for crystals of a finite size (σ) their melting temperature, i.e., σ is the length scale that controls the amount of melting point depression. All existing theoretical descriptions of these three phenomena yield a **temperature dependence** of this length scale “ ℓ ”.

The sentence: "More details are given in the main text." has been added to the caption of Figure 5 and Table 1.

Comment #11 of Reviewer #2: "On p. 2, we still read "thickness l of the wetting layer" while the Abstract described "molten regions of rather uniform width, W_{soft} " - how are l and W_{soft} different?"

The response says unconvincingly that one is the symbol for the experimental, the other for the theoretical thickness – where in science do we introduce different symbols like that?

Response of the authors: To avoid any misunderstanding, here we briefly and clearly point out the difference between ℓ and W_{soft} . In the context of surface premelting, ℓ is representing the theoretical thickness of the molten layer on a flat (semi-infinite) crystal with a smooth surface. W_{soft} is our experimentally measured width of the molten polyethylene layer at the meniscus region between the polyethylene crystal nanofilament and the mica substrate. However, caused by the difference in geometry and the additional interactions with the substrate, $W_{\text{soft}} \neq \ell$ but only proportional to ℓ , i.e., $W_{\text{soft}} \sim \ell$.

The first few sentences of the paragraph above Table 1 have been changed to:

When comparing $r^*(\Delta T)$, as given by equation (3), with the experimentally measured values of $W_{\text{soft}}(\Delta T)$, because we have to account for the specific geometry of the wedge-shaped soft zone of molten polymer and the influence of the mica substrate, $W_{\text{soft}}(\Delta T) \neq r^*(\Delta T)$, but only proportional to $r^*(\Delta T)$, i.e., $W_{\text{soft}}(\Delta T) \sim r^*(\Delta T)$. Thus, a constant prefactor C_{CNT} is required to relate $W_{\text{soft}}(\Delta T)$ with $r^*(\Delta T)$, we obtain $W_{\text{soft}}(\Delta T) = C_{\text{CNT}} \cdot r^*(\Delta T)$. Using $C_{\text{CNT}} = 16$ and employing equation (3) for $r^*(\Delta T)$ yields the red line shown in Figure 5c.

In the first paragraph after Table 1, the sentence has been changed to:

However, these approaches did not account for the influence of geometry, curvature, and the influence of the mica substrate, making a direct comparison with our results problematic.

Comment #12 of Reviewer #2: It still seems that generic l is introduced to hide the fact that sometimes it refers to a crystal thickness and sometimes to the thickness of a liquid wetting layer.

Response of the authors: In our responses to several previous comments given above in this reply, we have already explained how we related the crystal size to a thickness of a molten layer.

Comment #13 of Reviewer #2: If in the table instead of “ $l \sim$ ” the authors wrote “ $W_{\text{soft}} \sim$ ” followed by a crystallite thickness, the inconsistency would become apparent.

Response of the authors: The competition of interfacial energy and internal energy **RELATES** all three phenomena. The balance of an energy per surface (the “surface term”) and an energy per volume (the “volume term”), expressed by the ratio of “surface term”/“volume term”, introduces a **length scale** “ ℓ ”. The ratio of these two energies is at the basis of the size of the critical nucleus (r^*), reflects the thickness (l) of the liquid layer in surface premelting and determines for crystals of a finite size (σ) their melting temperature, *i.e.*, σ is the length scale that controls the amount of melting point depression. All existing theoretical descriptions of these three phenomena yield a **temperature dependence** of this length scale “ ℓ ”. In our manuscript, we have compared the predictions of the temperature dependence all three length scales (*i.e.*, $r^*(T)$, $l(T)$ and $\sigma(T)$) with our experimentally observed values of $W_{\text{soft}}(T)$. This comparison demonstrated that, within the uncertainty of the measured values of $W_{\text{soft}}(T)$, all three phenomena are compatible with our observations, *i.e.*, $W_{\text{soft}} \sim \ell$.

The end of the first paragraph of the introduction part has been changed to:

In the past, the size- and temperature-dependence of these phenomena have been well studied, showing that the physics behind these three phenomena originates from the broken symmetry of intermolecular interactions at interfaces, typically discussed by the balance of an energy per surface (the “surface term”) and an energy per volume (the “volume term”). The ratio of “surface term”/“volume term” introduces a length scale “ ℓ ” which reflects the key physical concept underlying all three phenomena.

Comment #14 of Reviewer #2: On p. 13, r^* is confusingly added to the mix of symbols, but it is treated as l in Table 1.

Response of the authors: Please see our answer to the previous comments.

Comment #15 of Reviewer #2: No answer was even attempted to my question regarding premelting: why is it favorable thermodynamically that at the amorphous-crystalline interface, part of the crystal is eaten away as temperature is raised, quite analogous to the observations in this manuscript? The long “response” simply goes off on tangents.

Response of the authors: In our previous reply, we have indicated that the thermodynamic analysis (*why is it favorable thermodynamically that at the amorphous-crystalline interface, part of the crystal is eaten away as temperature is raised*) is already done by many previous researchers. By citing their papers, we have written (on page 24 of our reply): *“As discussed in the above section, polymer chains within the nanofilament are almost exclusively oriented parallel to the filament axis. There is no surface covered with chain folds as it is exhibited in lamellar crystals of chain-folded polymers. Consequently, in the case of the investigated filaments, molten polymer chains are coexisting on the lateral surface of the crystalline core of the filaments, which may be identified as the “growth front” of the “oriented” polyethylene crystal, rather than the molten phase on the fold surface of lamellar polymer crystals, as reported by numerous other researchers, e.g., also by Tanabe, Strobl, and Fischer, **Surface melting in melt-crystallized linear polyethylene**, *Polymer*, 27, 1147-1153 (1986); [https://doi.org/10.1016/0032-3861\(86\)90001-7](https://doi.org/10.1016/0032-3861(86)90001-7) .”* Therefore, indeed *“part of the crystal is eaten away as temperature is raised”*.

There, we also pointed out the difference between the phenomenon reported in our manuscript and the complementary observations of melting of the fold surface of lamellar crystals, as described for example, in the paper from Tanabe, Strobl, and Fischer.

We have written (on Page 24 of our reply): *“We note that all concepts discussed above are based on thermodynamic parameters which apply to equilibrium situations. None of these concepts include any kinetic (time-dependent) aspects. Our results on $W_{\text{soft}}(T)$ show no dependence on the age of the sample, do not depend on filament size or morphology, and have been shown do be independent of molecular weight and dispersity of the polyethylene used. The interpretation of the*

results on surface melting by Tanabe et al. and later versions by M. Muthukumar[†] or by J.U. Sommer[‡] also rely on equilibrium, i.e., thermodynamic concepts.

*[†] Muthukumar, M. (2007). **Shifting Paradigms in Polymer Crystallization**. In: Reiter, G., Strobl, G.R. (eds) **Progress in Understanding of Polymer Crystallization**. Lecture Notes in Physics, vol 714. Springer, Berlin, Heidelberg. https://doi.org/10.1007/3-540-47307-6_1*

*[‡] Sommer, J.U. (2007). **Theoretical Aspects of the Equilibrium State of Chain Crystals**. In: Reiter, G., Strobl, G.R. (eds) **Progress in Understanding of Polymer Crystallization**. Lecture Notes in Physics, vol 714. Springer, Berlin, Heidelberg. https://doi.org/10.1007/3-540-47307-6_2*

Comment #16 of Reviewer #2: I had pointed out that the Introduction does not provide a motivation for this work. While the authors tell us that they “fully agree that providing a wide-ranging motivation at the beginning of the Introduction will significantly improve the manuscript by framing our work within a broader scientific context. Thus, we have revised the opening paragraph to articulate the fundamental questions and the broader implications of studying crystal-melt coexistence, the critical size required for (secondary) nucleation and its relation to surface melting and the melting point reduction in nanoscopic polymer filaments.” no motivation indicating the importance of this work beyond the “ivory tower” has been added. In a polymer journal this might be acceptable, but in a general journal like Nature Communications it is not.

Response of the authors: We are grateful to the reviewer for giving us the opportunity to re-emphasize the motivation of our comparison of our experimental observations with theoretical concepts nucleation, surface premelting and the melting point depression of finite size crystals. Our aim is to indicate there is a **RELATION** between these three phenomena. in order to avoid further confusion or misunderstandings, we now explicitly mention how the broken symmetry of intermolecular interactions at surfaces introduces a “common” length scale through the competition of interfacial energy (“surface term”) and internal energy (“volume term”). This description of the relation between the three phenomena has been added at the end of the first paragraph in the Introduction-section.

Comment #17 of Reviewer #2: I had asked if an observed very thin crystal (a bright region about 7 nm across) at high temperature is compatible with the Gibbs-Thompson equation, which tells us that such thin crystals melt away, in particular at high T . The authors acknowledged the presence of this thin crystal but did not answer my question. They deflected by referring to the filament thickness, but the Gibbs-Thompson equation is about the crystal thickness. This thickness needs to be accounted for quantitatively. Otherwise the claim of equilibrium in the manuscript title and elsewhere is in serious doubt.

Response of the authors: Given that the interplay of the acting intermolecular forces results in surface premelting (and not in nonmelting), a crystal, which has a thickness of *ca.* 7 nm at low temperatures, is obligatory covered by a liquid layer of thickness l when the temperature is increased towards the melting point. As shown, for example, in ref (1) of our manuscript, U. Tartaglino, T. Zykova-Timan, F. Ercolessi, E. Tosatti, **Melting and nonmelting of solid surfaces and nanosystems**, Phys. Rep. 411 (2005) 291–321, <https://doi.org/10.1016/j.physrep.2005.01.004>, the thickness $l(T)$ of this liquid layer can be related to the wetting behavior described via the corresponding surface and interfacial tensions. Depending on the type of interactions, the liquid surface layer may become detectable first at a temperature of about 3/4 of the bulk melting temperature. Thus, for our filaments of a width " w ", a crystalline core of $W_{\text{cryst}} = 7$ nm can only exist if $w \cong 2 \cdot W_{\text{soft}}(T) + 7$ nm. Assuming that w does not change with temperature T , the “bright region about 7 nm across” will be molten when $2 \cdot W_{\text{soft}}(T) \approx w$. This is predicted by the Gibbs-Thompson equation which, of course, relates a **length scale “ l ”** to the ratio of a “surface term” to a “volume term”. Therefore, to specifically answer the reviewer’s question: Yes, a bright region at high T is compatible as long as “ $2 \cdot W_{\text{soft}}(T) < w$. *In other words, when $2 \cdot W_{\text{soft}}(T) \approx w$, then the whole filament is molten.*”

Comment #18 of Reviewer #2: In summary, the manuscript remains deficient. It still requires major revisions before it can be considered for publication.

Response of the authors: We hope that we have removed any “doubts” the Reviewer # 2 might have had and that our manuscript can be considered for publication now.

AUTHORS' RESPONSES TO REVIEWERS' COMMENTS

Reviewer #2 (Remarks to the Author):

Comment #1 of Reviewer #2: I continue to perceive the authors' verbose responses mostly as obfuscation. If they contain valid arguments, then those should be added to the manuscript text.

Response of the authors: We tried our best to provide all required and valid arguments in our manuscript text. In addition, many supplementary details and profound in-depth discussions concerning the employed theoretical concepts were published in the cited references and therefore were not repeated in the manuscript text.

Comment #2 of Reviewer #2: For instance, I had asked how the presence of a very thin crystal (a bright region about 7 nm across) at high temperature is compatible with the Gibbs-Thompson equation, which tells us that such thin crystals melt away, in particular at high T. The authors have given a long response and I cannot tell if they're obfuscating or giving a deep explanation that says that the Gibbs-Thompson equation for the melting point of a thin crystal does not apply when the crystal is in a filament. They don't even refer to the Gibbs-Thompson equation at all. The response ends in: "Therefore, to specifically answer the reviewer's question: Yes, a bright region at high T is compatible as long as $2 \cdot W_{\text{soft}}(T) < w$. In other words, when $2 \cdot W_{\text{soft}}(T) \approx w$, then the whole filament is molten."

Response of the authors: We are sorry that Reviewer #2 is not satisfied with our answer. In Figure 5 c, we have referred to the Gibbs-Thomson equation, denoted by the black line. In addition, the two AFM images measured at 100 °C shown in the inset demonstrate that sufficiently small crystals are completely molten, while somewhat thicker crystals still remain, as predicted by the Gibbs-Thomson equation. In full agreement with the Gibbs-Thomson equation, we have provided a length-scale criterion for the transition from a crystal remaining to a crystal completely molten: *When $2 \cdot W_{\text{soft}}(T) \approx w$, then the whole filament is molten.*

Comment #3 of Reviewer #2: This deflection to "a bright region" rather than a thin crystal makes me suspicious that they may have been evading my question about the stability of a thin crystal at high temperature.

Response of the authors: In our response, we have adopted the term "*bright region*" from the previous Comment #17 of Reviewer #2, which reads: "*I had asked if an observed very thin crystal (a bright region about 7 nm across) at high temperature ...*". We believe that we have not evaded the reviewer's question as we showed two AFM images measured at 100 °C in the

inset of Figure 5 c, which demonstrate that sufficiently small crystals are completely molten, while somewhat thicker crystals still remain, as predicted by the Gibbs-Thomson equation.

Comment #4 of Reviewer #2: The authors also wrote in their response:

“Therefore, we believe a general critique of these fundamental theories is not justified. Regardless of the intended focus of the criticism, we maintain that our specific application of these theories to interpret our experimental data is both appropriate and insightful. We and the other two reviewers could not identify anything “sloppy and unconvincing” in these concepts and their theoretical descriptions.”

The first and last sentence is a quite obvious (and upsetting) misrepresentation of my evaluation. Quite clearly, my critique is not of the theories, but of their superficial presentation and sloppy and unconvincing application in this paper.

Response of the authors: The aim of this manuscript is not to discuss the three theoretical concepts, which are well known in physics and detailed in the cited references, but to show that the physics behind all three phenomena originates from the broken symmetry of intermolecular interactions at interfaces, reflected by the ratio of ⟨surface term/volume term⟩ which introduces a length scale “ ℓ ”. This has been done in a rigorous way.

Comment #5 of Reviewer #2: My conclusion remains that in this paper, talented experimentalists have recorded state-of-the-art images, and then someone added a superficial theoretical sauce mixing various concepts from the literature without a deeper understanding. The authors apparently lack the theoretical background to properly apply these theories, which they admitted in their earlier responses: “We agree that in our manuscript we only indicated/suggested the possibility of unifying various concepts, i.e., the concepts of nucleation, surface premelting, and finite-size melting point depression. Of course, a thorough proof of ****such unification requires a profound theoretical development, which is beyond the scope of this study.****”

Response of the authors: We have stated in our manuscript that we have compared the experimentally determined changes in the width with temperature, $W_{\text{soft}}(T)$, with the functional dependence on temperature (mostly given by a scaling law on the difference between melting temperature and sample temperature) predicted for melting point depression, (secondary) nucleation and surface premelting. We did not employ any prefactors predicted in the cited theoretical concepts as these values depend on sample geometry, interfacial curvature, and

substrate properties. For example, different results are obtained for differences in the force-distance dependence of the specifically acting intermolecular forces. To implement these specific factors, a more profound theoretical development is required, which is beyond the scope of this study.

Comment #6 of Reviewer #2: This limited theoretical understanding was also revealed by basic mistakes in their initial description of the quantities in their equations, which I pointed out and they conceded (which, of course, did not raise the level of the authors' theoretical understanding).

Response of the authors: We are glad that the reviewer agrees that in the revised manuscript we have provided the correct “*description of the quantities in the equations*”.

Comment #7 of Reviewer #2: Nevertheless, the authors keep stubbornly defending their superficial conflation of various theories, including their claim, in the Abstract, that they “*show* that the three phenomena are related”, with verbose and obfuscating responses but too few substantive changes in the text. They could easily have changed the wording to say that their analysis *suggests* that the phenomena are related. The theories even have different T-dependencies, see Table 1, and “fit” only because the experimental error margins are large, see Figure 5(b). I cannot support publication of such unconvincing work of doubtful quality.

Response of the authors: As mentioned above, we have employed the concepts of melting point depression, (secondary) nucleation and surface premelting as the underlying physics of all three phenomena originates from the broken symmetry of intermolecular interactions at interfaces, reflected by the ratio of ⟨surface term/volume term⟩ which introduces a length scale “ ℓ ”. We believe that Reviewer #2 agrees that, in this respect, the underlying physics is the same for all three phenomena. Therefore, and as shown in Figure 5 c, all three phenomena exhibit a similar functional dependence on temperature. To some extent, as shown for two theoretical predictions in the context surface premelting (blue lines in Figure 5c), functional dependences may differ for a specific distance-dependence of intermolecular interactions, e.g., screened Coulomb interactions (exponential function for the force distance dependence) or van der Waals interactions (often described by a Lennard-Jones power law potential). All this information had already been implemented in the revised version of our manuscript.

Review comments on NCOMMS-25-53009

The authors, based on intermittent AFM imaging, reported the coexistence of crystalline phase and molten phase of PE. The sample preparation and AFM imaging were carefully performed and therefore I can find some value in this manuscript, while I found some critical error in interpreting their AFM results. Thus, I cannot recommend for the manuscript in the current form to be published in this journal.

Judging from the temperature-dependent change in W_{soft} , the authors caught some realistic change of those soft molten regions. However, judging from force-distance curves in Fig. SI-4, I found that the situation is not so simple.

The above figure was clipped from their Fig. SI-4(a). They wrote that the curves 1 and 2 was obtained on the mica substrate, while those are not the shape of curves on pure mica. After the jump-in contact, there is a region with almost no change in force value (solid ellipse). This type of regions always observed when surface (or AFM probe itself) is contaminated. This contamination can be the molten state PE. It also can be adsorbed water molecules on mica. Or, something else. Or, the intercalation of solvent molecules happened during spin-coating. The probe surface can also be contaminated. The similar doubt was deduced from the withdrawing curve. The long tail (dashed ellipse) will never appear for the contact between pure mica and clean AFM tip. If so, what is phase zero in their intermittent AFM? The baseline is not any more the real baseline. Furthermore, these trends are also observed event at the

top of "hard" filaments. How they interpret it? Every discussion made in this manuscript need to be modified. Although the figure is not in the main text and therefore this is just a comment, but the vertical axis is somewhat strange. When I converted the force value into deflection value for curve 10 (assume that the spring constant is 5 N/m), the deflection difference between D of 5 nm and 10 nm is more than 5 nm. Thus, the sensitivity calibration was wrongly performed.

Second, I love their discussion using Fig. 2. Very careful. However, if looking at Figs. SI-4 and SI-7, soft layers are not symmetric especially when the scan direction is perpendicular to the direction of the filament. This is surely due to the tip-shape effect. I know this portion is not so easy, but the authors should mention something about the effect. If possible, they may be able to deconvolute the tip effect and once again discuss the value of W_{soft} .

Finally, this comment is not so critical but they should clearly describe the resonant frequency of the cantilever and actual vibration frequency. They may better to describe how to measure the resonant frequency. Just very far away from the sample surface like written in the manual of AFM or perform it at the vicinity of surface? I am asking this question because they used very small amplitude ratio. The resonant frequency changes a lot due to long-range

interaction force or air damping. If they were not so careful, all the measurement is not in "soft tapping" region but rather strong force is exerted on the specimen.

This paper provides convincing evidence for ~9-nm thin liquid layers along the edges of mica-supported semicylindrical polyethylene filaments, obtained by AFM, and tries to explain their temperature dependence. This is an intriguing system, which was apparently introduced earlier (see ref. 57 from 2015).

The title “Temperature dependence of crystal – melt coexistence for supported polyethylene filaments” does not describe the content of the paper accurately. (Below I’m proposing some experiments that could help fulfill the promise of this more ambitious title.) As it stands, “Temperature dependence of liquid edge layers of supported polyethylene filaments” would be an accurate title.

The theory presented centrally in this paper requires major revisions. Already in the Abstract, the authors name-drop “nucleation, surface premelting, and the melting-point depression of finite-size”, which are later confirmed to refer to the critical nucleus in classical nucleation theory (a lower limit of crystal size), premelting, and Gibbs-Thompson theory (another more stringent lower limit of crystal size). While at the end of Abstract and elsewhere, the authors claim to have unified these phenomena, the theory remains at a disappointingly low level, mostly just citing diverse sources (see Table 1 and the scant text below it). CNT and Gibbs-Thompson theories for *crystal size* are not really applicable to the thickness of a *liquid layer* studied here. Specifically, in the central Figure 5c, the authors use the two limiting *crystal* thicknesses to fit the *T*-dependence of a *liquid* layer. In the caption of Table 1, they make the misleading claim that the table shows “the characteristic length scale, represented as *l*, **of a liquid layer on the surface of a crystal**, as predicted by three different phenomena”, while at least two of the entries are actually the limiting thicknesses of *crystals*. On p. 2, we read “thickness *l* of the wetting layer” while the Abstract described “molten regions of rather uniform width, W_{soft} ” - how are *l* and W_{soft} different? It seems that generic *l* is introduced to hide the fact that sometimes it refers to a crystal thickness and sometimes to the thickness of a liquid wetting layer. If in the table instead of “*l* ~” the authors wrote “ W_{soft} ~” followed by a crystallite thickness, the inconsistency would become apparent. On p. 13, r^* is confusingly added to the mix of symbols, but it is treated as *l* in Table 1.

The theory on p. 13 contains technical flaws in the definition of the quantities (see details below), which suggests that the authors were struggling with the concepts. It seems that the simple theory on p. 13 is suitable for *primary* nucleation, but the authors make it clear that they want to consider *secondary* nucleation. In secondary nucleation, there should be two characteristic length scales, along the preexisting surface and perpendicular to it, not just a single r^* . I see this confirmed in the corresponding author’s cartoon of a secondary critical nucleus in reference 50, which is one chain stem thick and five stems wide. The assumption of a cylindrical secondary nucleus (top of p. 12) is unreasonable. And why can’t the existing fibrillar crystal grow without nucleation, as in Regime I of Lauritzen-Hoffman theory?

What the data do seem to show is premelting. It is included in Table 1 but not really discussed. It needs to be explained that while the crystal is the lowest free-energy form of the polymer, surface energy of the air-polymer interface makes insertion of a liquid-like layer thermodynamically favorable. It should also be discussed that in bulk polymers, premelting has been invoked, e.g. by Tanabe, Strobl, and Fischer ([https://doi.org/10.1016/0032-3861\(86\)90001-7](https://doi.org/10.1016/0032-3861(86)90001-7)) in studies of HDPE, with a different meaning: without air, at the amorphous-crystalline interface, eating away part of the crystal as temperature is raised, quite analogous to the observations in this manuscript – why is this favorable thermodynamically?

The literature on conventional premelting tells us that any crystal, in particular near its melting point, is covered with a liquid-like layer. Why is there no such liquid layer on top of the crystalline filament in Figure 3?

One could envision that entanglements that have been excluded from the growing crystal and enriched outside the crystal prevent crystallization of the liquid layer. This is invoked by some experts to explain lamellar crystallites alternating with amorphous layers in semicrystalline polymers. Is this entanglement mechanism excluded for the liquid layers observed in this study?

On p. 2, at the very beginning of the Introduction, the authors do not present a motivation for their work. I do find it intrinsically interesting, but it would be better if a motivation or grander framework was included here.

Figure 5c displays intriguing images that raise interesting questions. The image at top shows a very thin crystal (the bright region about 7 nm across) at high temperature – is such a small thickness compatible with the Gibbs-Thompson equation, which tells us that thin crystals melt away, in particular at high T ? Have the authors raised temperature and seen the crystal melt suddenly when its thickness drops below the Gibbs-Thompson stability limit?

At the bottom of the figure, we see a completely melted filament. Observing it at gradually decreasing temperature when at some point a critical nucleus must form would seem like a worthwhile investigation, but the authors do not seem to consider critical nuclei in this relevant context.

I'm quite uncomfortable with the way in which the authors use the term "polymers" (which they do a lot, 50 times in total). To me, examples of polymers are PE, iPP, and PET. The authors use "polymers" where I would prefer "polymer chains", "polymer segments", or "polymer".

Examples:

P. 3, center: "the limited number of polymers available" should probably be changed to "the limited number of polymer chains available".

P. 3, 5th line from bottom: "interactions between stretched polymers," should probably be changed to "interactions between stretched polymer chains,".

P. 4, first paragraph: "filaments of crystalline chains were surrounded initially by non-stretched molten polymers." should probably be changed to "filaments of crystalline chains were surrounded initially by non-stretched molten polymer segments."

P. 4, 2nd paragraph: "the number of available polymers is above a threshold value." Should be changed to "the number of available polymer chains [or segments?] is above a threshold value."

P. 4, below center: "sufficient mobility of noncrystallized polymers but also" should be changed to "sufficient mobility of noncrystallized polymer but also" or "sufficient mobility of noncrystallized polymer segments but also"

P. 5, second sentence: "contained molten polymers only"

Etc. etc., including

P. 18, line 7/8: "contained molten polymers which bounded the solid core region of the filament consisting of crystallized polymers." should be changed to "contained molten polymer which bounded the solid core region of the filament consisting of crystallized polymer."

The text contains various other flaws; here are some examples:

P. 1, first line of the Abstract: “is causing that crystals are often” Please rephrase. E.g. “results in crystals often being”

P. 2 below center: “several theoretical approaches” could the authors name some of them here?

P. 2 last paragraph, “for a finite number N of molecules and a finite volume V ”: I believe the authors mean “for a fixed number N of molecules and a fixed volume V ”

P. 3 near the top, “and T are constant”: It would be better to add “while the particle number N can vary/fluctuate” for clarity.

P. 3 second paragraph: “In the here presented study” needs to be changed to “In the study presented here”. Similarly for “here applied conditions” on p. 7, “The here investigated filaments” on p. 8, and “here investigated filaments” on p. 10.

P. 3 in the 2nd paragraph, “polymers in ultrathin layers ... crystallized very slowly.” I think this should be changed to present tense: , “polymers in ultrathin layers ... crystallize very slowly.”

P. 5, third sentence starting with “Naively”. It should be made clear that the “uniform mechanical properties” should be considered across rather than along the filaments.

P. 5, captions of Figure 5: “or a-PE-B”.

P. 5, 2nd line from the bottom can be made clearer: “For ~~an~~ easier comparison of a) with b) and c) with d),”

P. 6: “in the MethodS” section.”

P. 8 right above Figure 3: “coexisted with molten EDGES, soft zones at the LATERAL boundaries of the filament.”

P. 9: The tense of verbs in the first paragraph should be changed to present tense.

P. 9, beginning of second paragraph: “SOME with a uniform width and OTHERS decorated”.

P. 13 top: What is the basis of “the assumption of $W_{\text{soft}} \sim r^*$ ”?

P. 13, 4th line: “polymer is not expected to influence neither r^* nor W_{soft} .” contains too many negatives. Correct: “polymer is expected to influence neither r^* nor W_{soft} .” or “polymer is not expected to influence either r^* or W_{soft} .”

P. 13, below eq.(1): “ ΔG_V is the free energy difference involved in the formation of a crystal nucleus”. Dimensional analysis indicates that “ ΔG_V is the free energy change, per volume, ” of the formation of a crystal nucleus”.

P. 13 below eq.(2): “ ρ_n is the number density of molecules” is incompatible with later “ ρ_n is ranging from ca. 780 kg/m³ to ca. 970 kg/m³”, which shows that “ ρ_n is the mass density of the nucleus”.

P. 13 below eq.(2): “ ΔH_f is the heat of fusion per volume element” is incompatible with “ $\Delta H_f = 4.1 \pm 0.2$ kJ/mol of CH₂”. Dimensional analysis indicates that): “ ΔH_f is the heat of fusion per mass” or “ ΔH_f is the specific heat of fusion”.

P. 13: I would propose to write “nm” at the end of the equation instead of the awkward “in [nm]”.

P. 14, “on various filaments, either they had a”: Better: “on various filaments, with either a”

P. 15, Figure 5: In part c), it would be good to mark the absolute T values on the top axis, to counter the impression that W decreases with increasing temperature.

P. 15, Figure 5: The font sizes are too disparate. They should be made more uniform (mostly by increasing the size of the small lettering).

P. 15, caption of Figure 5: The sentence starting with “The two AFM phase images” seems to lack a main verb.

P. 16, right below the table: “and possible intercepts.” should probably be rephrased as “and additive constants.”

P. 17: “Spin coating experimentS were performed”